# Enabling programmable dynamic DNA chemistry using small-molecule DNA binders

Junpeng Xu[1,2], Guan Alex Wang[1], Lu Gao[1], Lang Wu[1], Qian Lei[3], Hui Deng [3] & Feng Li [1,2,4] ✉

The binding of small molecules to the double helical structure of DNA, through either intercalation or minor groove binding, may significantly alter the stability and functionality of DNA, which is a fundamental basis for many therapeutic and sensing applications. Here, we report that small-molecule DNA binders can also be used to program reaction pathways of a dynamic DNA reaction, where DNA strand displacement can be tuned quantitatively according to the affinity, charge, and concentrations of a given DNA binder. The binder-induced nucleic acid strand displacement (BIND) thus enables innovative technologies to accelerate the discovery and characterization of bioactive small molecules. Specifically, we demonstrate the comprehensive characterization of existing and newly discovered DNA binders, where critical parameters for binding affinity and sequence selectivity can be obtained in a single, unbiased molecular platform without the need for any specialized equipment. We also engineer a tandem BIND system as a high-throughput screening assay for discovering DNA binders, through which 8 DNA binders were successfully discovered from a library of 700 compounds.

Interactions between small-molecule binders and double-stranded DNA (dsDNA) are the fundamental basis of many anticancer and anti-infection pharmaceuticals[1–4] and of chemical probes for sensing, imaging, and quantifying nucleic acids in diverse settings[5–9]. Recent advances in designing, screening, and characterizing DNA binders have not only promoted our understanding of their essential roles in critical biological processes, such as DNA replication[10], DNA damage[11,12], and transcriptional regulation[13], but also enabled exciting applications ranging from guiding the differentiation of pluripotent stem cells[14] to epigenetic regulation[15] and to inducing trinucleotide-repeat contraction[16]. Beyond their important roles in biological systems, DNA binders have also found increasing applications in artificial systems for constructing and controlling DNA origami assembly and conformation[17], stabilizing DNA nanostructures against low-magnesium buffers and nuclease degradation[18,19], modulating the bond strength of DNA-nanoparticle superlattices[20], and constructing DNA-based rectifiers and excitonic circuits[21,22]. To date, most applications of DNA binders rely on their static interactions with either dsDNA or more complex DNA structures through intercalation or minor groove binding.

First introduced by Yurke et al. in 2000, toehold-mediated DNA strand displacement (TMSD) has become the most widely used elementary reaction in dynamic DNA nanotechnology[23,24]. Mechanistically, TMSD can be considered an $S_N2$ reaction, which is initiated at a short complementary single-stranded DNA (ssDNA) toehold domain and progresses through a branch migration process to displace a prehybridized DNA strand. In this reaction pathway, metal cations, such as $Mg^{2+}$, play critical roles in stabilizing the initial duplex as well as the triplex intermediate against the repulsion caused by the dense negative charges of phosphorate backbones (Fig. 1b)[25]. By placing

[1]Key Laboratory of Green Chemistry and Technology of Ministry of Education, College of Chemistry, Sichuan University, Chengdu, Sichuan 610064, P. R. China. [2]Department of Chemistry, Centre for Biotechnology, Brock University, St. Catharines, ON L2S 3A1, Canada. [3]Department of Respiratory and Critical Care Medicine, Institute of Respiratory Health, Targeted Tracer Research and Development Laboratory, West China Hospital, Sichuan University, Chengdu, Sichuan 610064, P. R. China. [4]Med+X Center for Manufacturing, West China Hospital, Sichuan University, Chengdu, Sichuan 610041, P. R. China. ✉e-mail: fli-scu@outlook.com

TMSD in a solution containing no metal cations or very low concentrations of monovalent metal cations (<25 mM), it is possible to switch the dominant strand displacement reaction pathway to $S_N1$, where the duplex is first dissociated because of the strong charge repulsion and then exchanged with the invader strand (Fig. 1c)[26].

In this work, we demonstrate that reaction pathways of TMSD under low metal cation conditions can be further programmed using small molecular DNA binders. We name this binder-responsive dynamic DNA reaction as binder-induced nucleic acid strand displacement (BIND). Specifically, the addition of low concentrations of dsDNA binders suppresses the $S_N1$ reaction pathway by stabilizing the duplex reactant through specific DNA-binder interactions and thus reduces the yield of the overall strand displacement in a concentration-dependent manner (Fig. 1d). Upon fully occupying the initial prehybridized DNA duplex, free DNA binders then facilitate the docking of the toehold domain and thus promote stand displacement via an $S_N2$ reaction pathway (Fig. 2e). More importantly, the combined experimental measurement and theoretical modeling reveal that the Gibbs free energies of BIND at both $S_N1$ and $S_N2$ regions depend quantitatively on the binding affinity, charge condition, and concentrations of a given DNA binder, allowing precise programming of the strand displacement pathways. Based on the quantitative relationship, we further engineered BIND into a molecular tool for the one-stop comprehensive profiling of small-molecule binders, where critical binding parameters, including binding constant ($K_d$), binding site size ($n$), cooperativity ($\omega$), enthalpic ($\Delta H$) and entropic ($-T\Delta S$) contributions, as well as sequence selectivity, can be accurately determined without the need for any specialized equipment. We also introduce a tandem BIND reaction system and engineer it into an HTS assay, through which eight new DNA binders were successfully discovered from a library of 700 compounds.

## Results

### BIND mechanism

BIND is a DNA strand displacement reaction with programmable reaction pathways in response to small-molecule DNA binders. A typical BIND reaction was initiated by co-incubating 20 nM DNA duplex (CP) with a certain DNA binder for 5 min, followed by a strand displacement reaction driven by an invader strand I (Supplementary Fig. 1). Because the prehybridized CP is destabilized by the strong charge repulsion, the displacement between CP and I involves the dissociation of CP (reaction 1) and subsequent hybridization between C and I (reaction 2), corresponding to a classic $S_N1$ reaction pathway (Fig. 1c):

$$CP \rightarrow C + P \qquad (1)$$

$$C + I \rightarrow CI \qquad (2)$$

$$CP + I \rightarrow CI + P \qquad (3)$$

In our study, the equilibrium constant in the absence of any binder was determined experimentally for a representative strand displacement reaction (Supplementary Table S1) in a Tris-EDTA (TE) buffer containing no metal cations. The initial reaction free energy $\Delta G_{ini}^\circ$ of the strand displacement was then determined to be 0.33 kcal/mol. Upon the addition of SYBR Green I (SG-I, $K_d = 7.1$ nM) as a model DNA binder, we observed sharp decreases in reaction yields against increasing concentrations of SG-I until a critical binder concentration (CBC) at which the CP duplex was fully occupied (Fig. 2a and Supplementary Fig. 2). By changing the length of the CP duplex, we also found that CBC shifted to higher concentrations upon increasing the CP length. Nevertheless, the ratio between CPC and duplex length was kept a constant (Supplementary Fig. 3). Upon further raising the

concentration of SG-I above CBC, we observed the acceleration of strand displacement through an $S_N2$ reaction pathway (Fig. 2a and Supplementary Figs. 2–3):

$$CP + I \rightarrow CPI \rightarrow CI + P \qquad (4)$$

This observation suggests that high concentrations of free SG-I could facilitate toehold docking, which initiates the process of branch migration and strand displacement. The proposed dissociative ($S_N1$) and displacement ($S_N2$) reaction pathways were further confirmed using kinetic analysis (Supplementary Figs. 4–7).

To quantitatively understand how SG-I programs the BIND reaction, we next developed a theoretical model that considers both $S_N1$ and $S_N2$ reaction pathways for strand displacement (detailed in Supplementary Information Section S2 and Supplementary Fig. 8). We then fit the experimental data in Fig. 2a into the model to determine the quantitative relationship between SG-I concentration and the overall equilibrium constant $K_{BIND}$ (Fig. 2b) as well as the reaction free energy $\Delta G_{BIND}^\circ$ (Fig. 2c). Interestingly, SG-I was found to program $\Delta G_{BIND}^\circ$ in a linear fashion at both $S_N1$ and $S_N2$ regions (Fig. 2c). We further confirmed that this quantitative relationship was generalizable to other DNA binders regardless of their binding modes to DNA (intercalator or groove binding) (Supplementary Fig. 9). $\Delta G_{BIND}^\circ$ can thus be expressed in the following equations, and BIND becomes predictable at any given binder concentration:

$$\Delta G_{BIND}^\circ = \Delta G_{ini}^\circ + \theta \cdot C_{Binder}, (C_{Binder} \leq CBC, SN1\ reaction\ pathway) \qquad (5)$$

$$\Delta G_{BIND}^\circ = \Delta G_{ini}^\circ + (\theta - \xi) \cdot CBC + \xi \cdot C_{Binder}, (C_{Binder} > CBC, SN2\ reaction\ pathway) \qquad (6)$$

where $\Delta G_{BIND}^\circ$ is the reaction free energy of the overall BIND reaction, $\Delta G_{ini}^\circ$ is the initial reaction free energy in the absence of DNA binders, $C_{Binder}$ is the concentration of DNA binder, $\theta$ is an activity coefficient for inhibiting strand displacement, and $\xi$ is an activity coefficient for promoting strand displacement. Both $\theta$ and $\xi$ are intrinsic properties of a given DNA binder and can be determined quantitatively as the slopes at the $S_N1$ pathway region and $S_N2$ pathway region, respectively.

By plotting $\theta$ and $\xi$ against binders with increasing binding affinities, we found that $\theta$ was determined solely by the binding strength of the DNA binder (Fig. 2d), whereas $\xi$ was determined by both the binding affinity and the charge condition (Fig. 2e). For example, echinomycin is a much stronger binder than Ru(Phen)$_3$Cl$_2$ but showed much less promotion of strand displacement in the $S_N2$ pathway because it is a neutral binder. Using 1-benzylquinolinium chloride (1-BQC) as a positively charged nonbinder, we also confirmed that BIND is highly binder-specific in both the $S_N1$ and $S_N2$ reaction pathways. Therefore, four classes of distinct BIND profiles can be established for small molecules with varying combinations of affinities and charge conditions: strong positively charged binders ($\theta > 0$, $\xi < 0$) (Fig. 2f), strong neutral binders ($\theta > 0$, $\xi = 0$) (Fig. 2g), weak positively charged binders ($\theta = 0$, $\xi < 0$) (Fig. 2h), and nonbinders ($\theta = 0$, $\xi = 0$) (Fig. 2i). Here, we consider binders with $K_d$ values below 1 µM to be a strong binder. The theoretically predicted profiles were further confirmed experimentally using representative compounds, including Berenil (Fig. 2j), echinomycin (Fig. 2k), Ru(Phen)3Cl$_2$ (Fig. 2l), and 1-BCQ (Fig. 2m).

### Profiling DNA-binder interactions using BIND

Having found that the $S_N1$ region of BIND was quantitatively determined by the affinities of small-molecule DNA binders, we next employed BIND to quantitatively profile the thermodynamics of DNA-binder interactions. Because the fluorescence signal is generated by

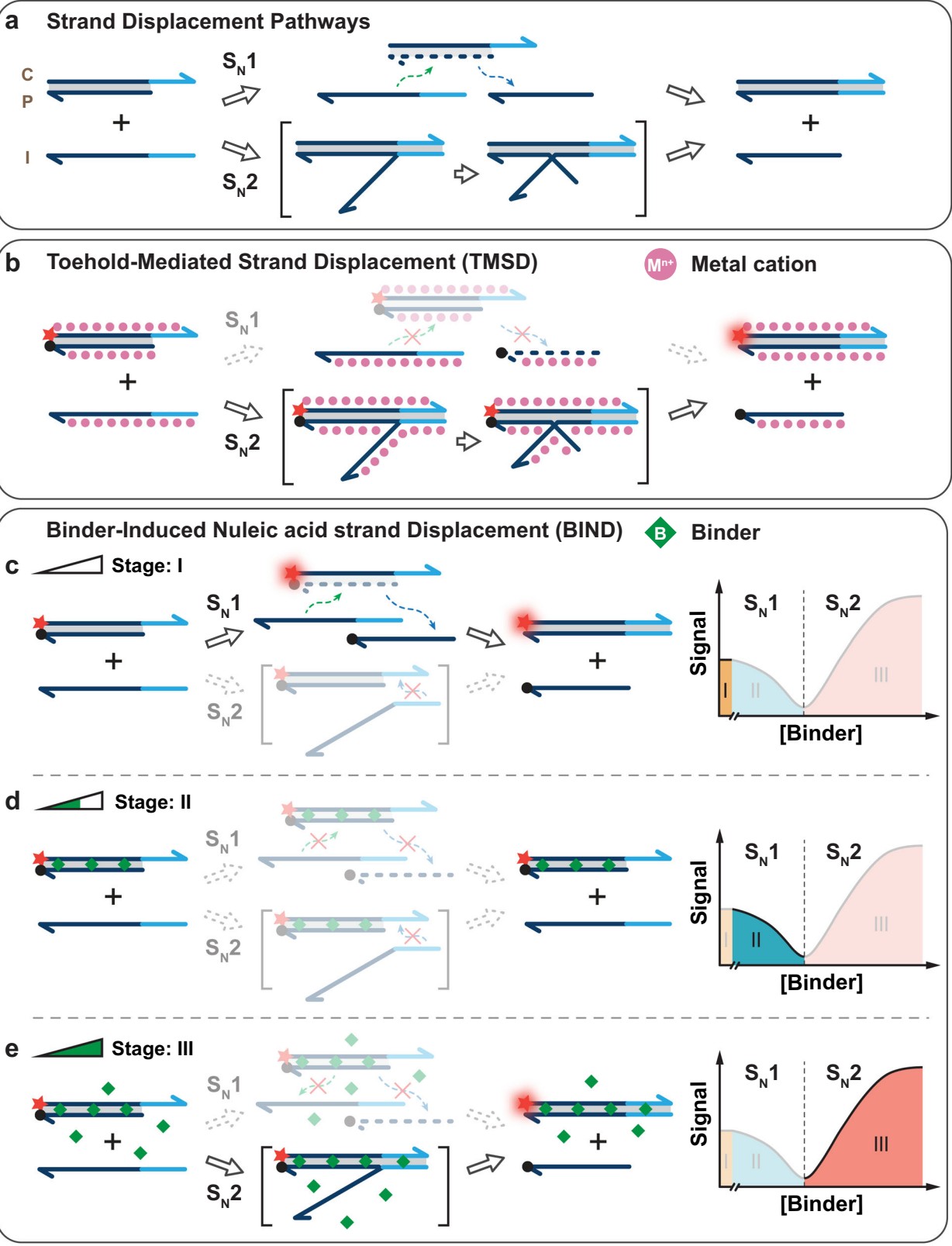

**Fig. 1 | Reaction pathways for DNA strand displacement reactions. a** Schematic illustration of two possible reaction pathways ($S_N1$ and $S_N2$) for a typical toehold-mediated strand displacement reaction. **b** The conventional strand displacement reaction is dominated by the $S_N2$ reaction pathway, where metal cations help stabilize the triplex intermediate. **c** Activation of the $S_N1$ reaction pathway by preparing the displacement reaction in a buffer containing no or very low concentrations of metal ions, which is labeled as domain I in the BIND curve. **d** Inhibition of both reaction pathways in response to low concentrations of DNA binders in BIND, which is labeled as domain II in the BIND curve. **e** Activation of the $S_N2$ reaction pathway in response to high concentrations of DNA binders in BIND, which is labeled as domain III in the BIND curve.

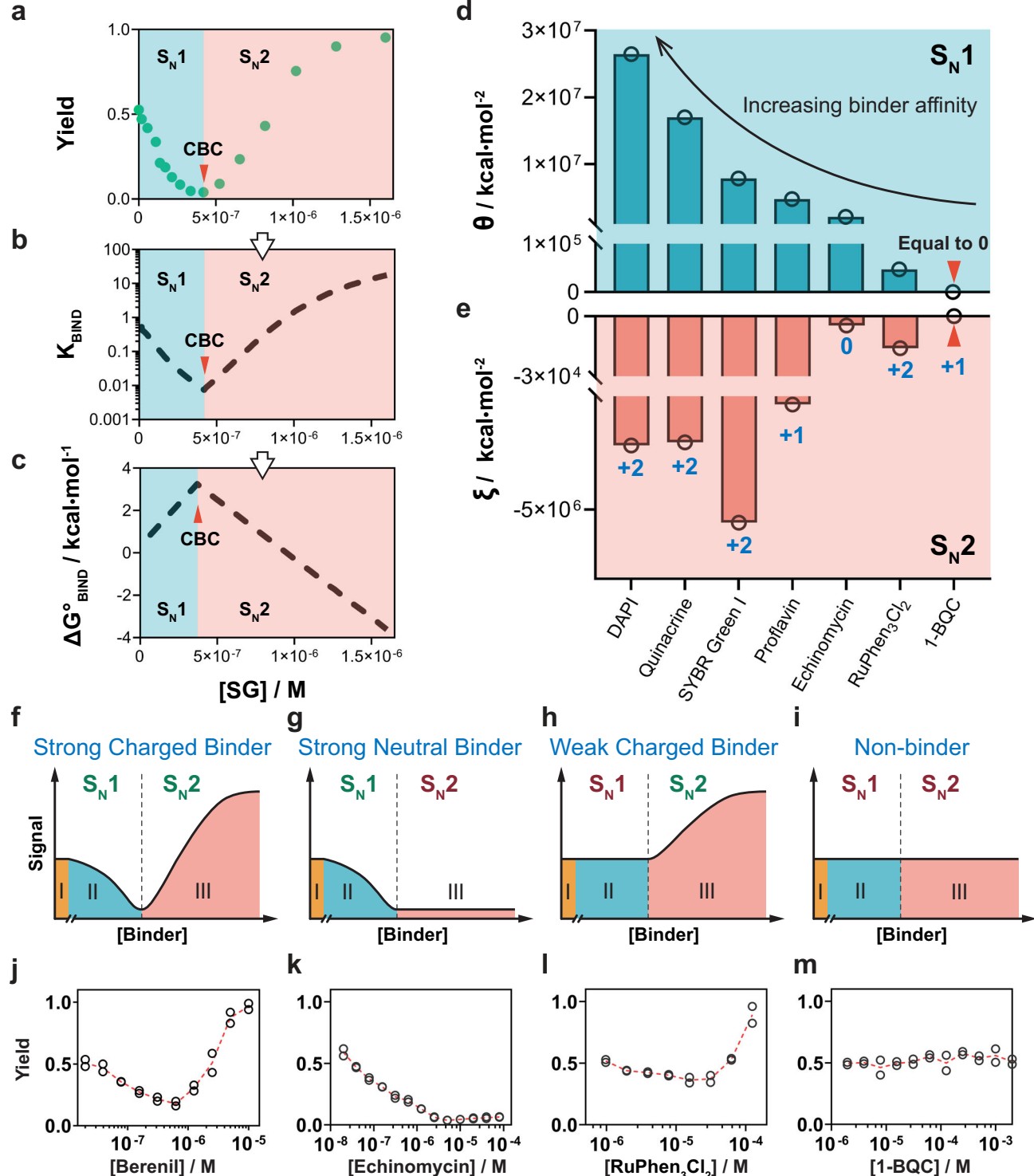

**Fig. 2 | BIND profiles in response to binder affinity and charge condition. a** A typical BIND curve established by plotting strand displacement reaction yields against the concentrations of SG-I. **b** Quantitative relationship between $K_{BIND}$ and SG-I concentration achieved by fitting experimental data into a theoretical model of BIND. **c** A further mathematical conversion revealed the linear relationship between $\Delta G_{BIND}$ and SG-I concentration. **d, e** Changes in activity coefficients **θ** at the $S_N1$ region (**d**) and **ξ** at the $S_N2$ region (**e**) in response to binders with varying binding affinities and charge conditions. **f–i** Schematic illustration of four distinct BIND profiles against positively charged strong binders (**f**), neutral strong binders (**g**),

positively charged weak binders (**h**), and nonbinders (**i**). Domains I, II, III represent the BIND in the absence of binder, binder concentration below CBC, and binder concentration above CBC, respectively. **j** Experimentally established BIND profile for a representative positively charged strong binder, SG-I. **k** Experimentally established BIND profile for a neutral strong binder, echinomycin. **l** Experimentally established BIND profile for a representative positively charged weak binder, RuPhen₃Cl₃. **m** Experimentally established BIND profile for a representative positively charged nonbinder, 1-BQC.

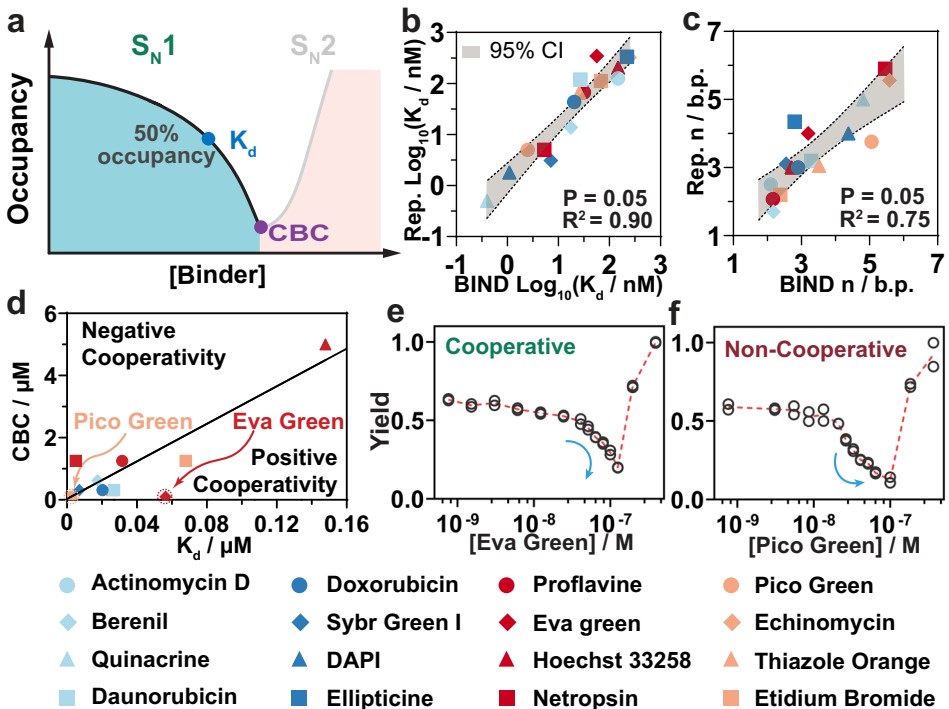

**Fig. 3 | Profiling affinities of DNA-binder interactions using BIND. a** Schematic illustration of determining the binding affinities of small-molecule DNA binders using the $S_N1$ domain of the BIND curve. **b** $K_d$ values of 16 binders determined using BIND, which were then plotted against the reported values in the literature. **c** Binding site sizes of 16 binders determined using BIND, which were then plotted against the reported values in the literature. Pearson correlation analysis was used to determine the confidence interval with $P = 0.05$ ($n = 12$), which were highlighted in gray shading. **d** Profiling the binding cooperativity of 16 binders by plotting their critical binder concentration (CBC) against the $K_d$ values determined using BIND. **e** BIND profile of the cooperative binder EVA Green. **f** BIND profile of a non-cooperative binder Pico Green.

externally labeled fluorophore-quencher pairs and strand displacement reactions, BIND does not rely on the intrinsic fluorescent properties of DNA binders and is thus a universal assay generalizable to all DNA binders (Supplementary Fig. 10). Experimentally, the binding isotherm was established by plotting the attenuated fluorescent signals against binder concentrations at the region of the $S_N1$ reaction pathway, where 0% and 100% occupancies were established at binder concentrations of zero and CBC, respectively (Fig. 3a). Both binding affinities ($K_d$) and the binding site size ($n$) were determined by fitting the binding curves at the $S_N1$ region against the lattice-ligand model developed by McGhee and Von Hippel (Supplementary Fig. 11).

A head-to-head comparison between BIND and the classic fluorescence turn-on assay for measuring the binding affinities of SG-I to dsDNA revealed highly consistent values for both $K_d$ (7.0 nM vs. 9.3 nM) and $n$ (2.6 vs. 2.4 nM), confirming the feasibility of BIND for profiling the binding properties of small-molecule DNA binders (Supplementary Fig. 12). We further expanded the validation of BIND against 16 known DNA binders with reported $K_d$ values ranging from 0.4 nM to 250 μM (Supplementary Fig. 13). Values of $K_d$ and $n$ were successfully determined for all 16 binders using BIND, regardless of the binding mode (7 intercalators, 8 groove binders, and 1 unknown mode) and charge condition (13 positively charged binders and 3 neutral binders), and the results were highly consistent with reported values determined under similar conditions (Fig. 3b, c, and Supplementary Table S2).

In addition to $K_d$ and $n$, binding cooperativity ($\omega$) is another important parameter to characterize how the binding of one binder affects the binding of the next. A binder of positive cooperativity is likely to attract free binders to fill the empty neighboring slots on the dsDNA, which is featured by a sharp binding curve with a narrower dynamic range than that of a noncooperative binder. Interestingly, when CBC was plotted against the $K_d$ values of the 16 binders, all

noncooperative binders formed a line, as shown in Fig. 3d, which separates the quadrant into two parts. Binders distributed in the lower region had a shift of $K_d$ to a higher numeric value and thus are positively cooperative. To test this hypothesis, we measured $\omega$ for Pico Green and Eva Green, which were clearly distributed in the space of noncooperative binders (on the line) and that of positively cooperative binders (in the lower space) in Fig. 3d, respectively. Despite the same CBC at 100 nM, the $K_d$ of Eva Green was 25 times greater than that of Pico Green (Fig. 3e, f). We further fit the binding curve established using BIND to the mathematical model previously developed by McGhee and Hippel (detailed in the Methods section)[27]. The cooperativity of Eva Green was determined to be 13.2, suggesting that the binding of Eva Green was highly positively cooperative, which is consistent with a previous study by Shoute and Loppnow ($\omega = 8.1$)[28].

## Profiling thermodynamics of DNA-binder interactions using BIND

When employed for profiling the binding characteristics of DNA binders, BIND was found to be highly sensitive and reproducible, and capable of accurately resolving binding constants at temperature intervals <2 °C. This feature allows us to gain critical thermodynamic parameters of DNA-binder interactions, including free energy ($\Delta G$), enthalpy ($\Delta H$), and entropy ($\Delta S$), without the need to melt the DNA over wide temperature ranges. We measured the $K_d$ values of 15 binders using BIND at 27 °C, 30 °C, 32 °C, 35 °C, and 37 °C and fit the results using Van't Hoff's equation (Supplementary Figs. 14–29). We ranked 15 binders according to the measured $\Delta S$ values at 37 °C in Fig. 4a, where $\Delta G$ ranges from −13.3 to −9.4 kcal/mol, $\Delta H$ ranges from −14.3 to +1.9 kcal/mol, and $-T\Delta S$ ranges from −12.7 to +3.5 kcal/mol at 37 °C. We also calculated the enthalpy contribution to the overall Gibbs free energy ($\Delta H/\Delta G$) for 12 binders with reported thermodynamic parameters ranging from −0.2 to +1.2 (Fig. 4b). The enthalpy

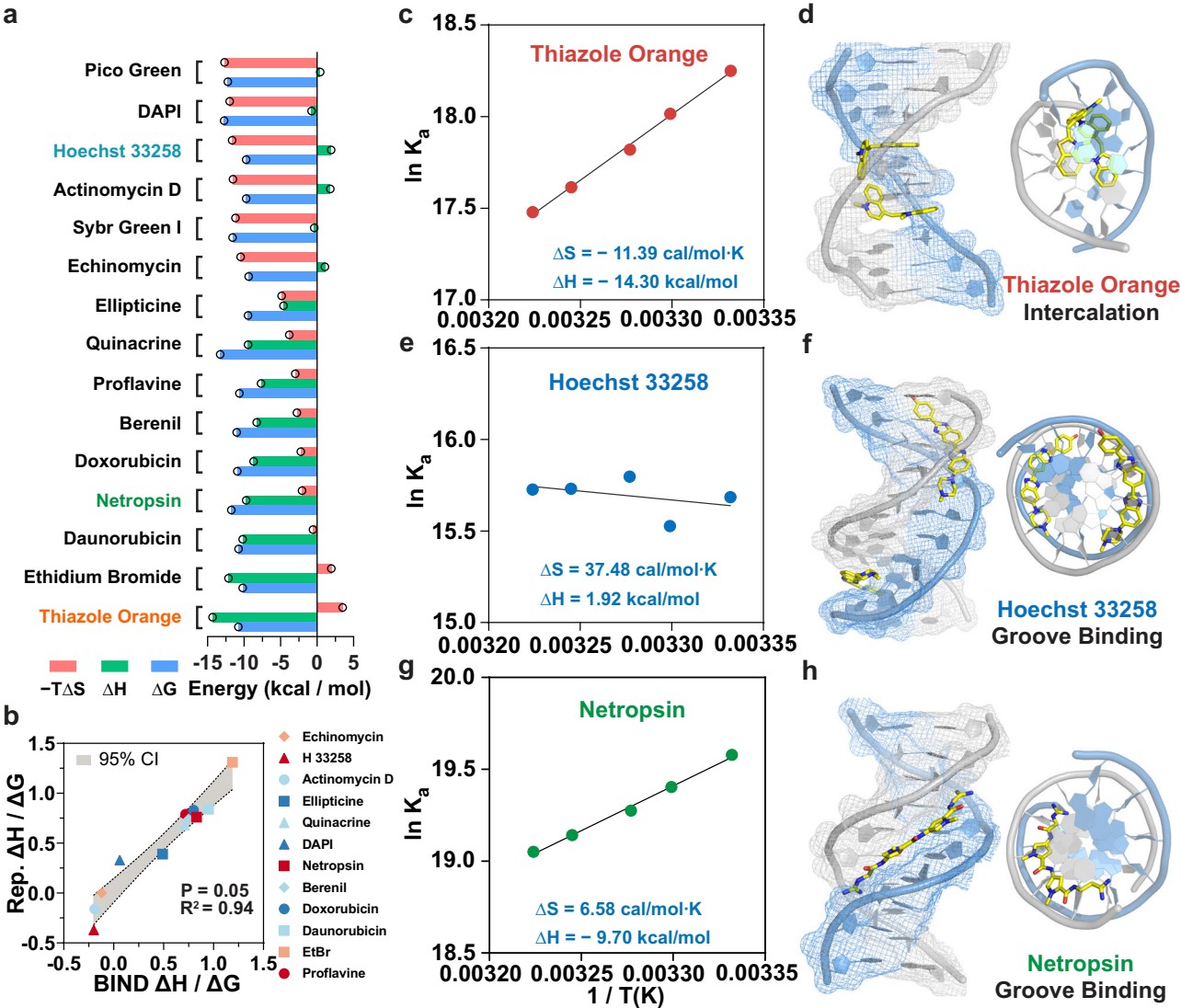

**Fig. 4 | Profiling the thermodynamics of DNA-binder interactions using BIND.**
**a** Numerical values of ΔG, ΔH, and −TΔS for 15 DNA binders using BIND. These values were ranked against −TΔS. **b** Comparing values of enthalpy contribution obtained using BIND and values reported in the literature for 12 binders with previously reported thermodynamic parameters. Pearson correlation analysis was used to determine the confidence interval with $P = 0.05$ ($n = 12$), which was highlighted in gray shading. **c** Determining ΔH and ΔS of a representative intercalator, thiazole orange, by plotting ln$K_a$ against reaction temperature and fitting using Van't Hoff's equation. **d** Predicted binding mode of thiazole orange using molecular docking. **e** Determining ΔH and ΔS of a representative minor groove binder, Hoechst 33258, by plotting ln$K_a$ against reaction temperature and fitting using Van't Hoff's equation. **f** Predicted binding mode of Hoechst 33258 using molecular docking. **g** Determining ΔH and ΔS of netropsin by plotting ln$K_a$ against reaction temperature and fitting using Van't Hoff's equation. **h** Predicted binding mode of netropsin using molecular docking.

contributions measured using BIND were highly consistent with reported values with $R^2 = 0.94$ (Fig. 4b and Supplementary Table S3).

Analysis of the enthalpic and entropic contribution to the free energy of DNA binder interactions can reveal the molecular forces that drive the binding and thus shed light on the binding mode of each binder[29]. For example, BIND revealed a favorable enthalpic contribution of −14.3 kcal/mol for a classic intercalator, thiazole orange, indicating that the main driving force for binding is base stacking and the formation of hydrogen bonds (Fig. 4c). The entropic penalty of +3.5 kcal/mol also indicated the loss of rotational degrees of freedom, likely due to the constraint structure caused by the intercalation (Fig. 4d). While intercalation is mainly enthalpically driven, the driving force for the minor grooving binding can be highly diverse[29]. Using BIND, we demonstrated that the binding of Hoechst 33258, a classic minor groove binder, was entropically driven via hydrophobic and electrostatic interactions, evidenced by a favorable entropy

contribution (−TΔS) of −11.6 kcal/mol with a slight enthalpic penalty of +1.9 kcal/mol (Fig. 4e, f). In contrast, as an amide-linked minor groove binder, netropsin generated a favorable enthalpy contribution of −9.7 kcal/mol with a slight contribution from entropy (−2.0 kcal/mol) upon binding to DNA, suggesting that the binding was driven by hydrogen bonding and/or van der Waals interactions (Fig. 4g, h)[30].

**Determining the sequence selectivity of DNA binders using BIND**
DNA binders with high sequence selectivity can serve as antagonists of transcription factors or inhibitors of gene expression and thus hold great therapeutic potential. Therefore, we next engineered BIND to evaluate the sequence selectivity of DNA binders in terms of their preference for GC- or AT-rich sequences. To achieve this goal, we designed a panel of stem-looped sink probes, each containing an 8 bp stem in the format of 5'-CGXXXXXC-3' and a 5-dA loop (Fig. 5a). DNA binders were incubated with a mixture of a sink probe containing a

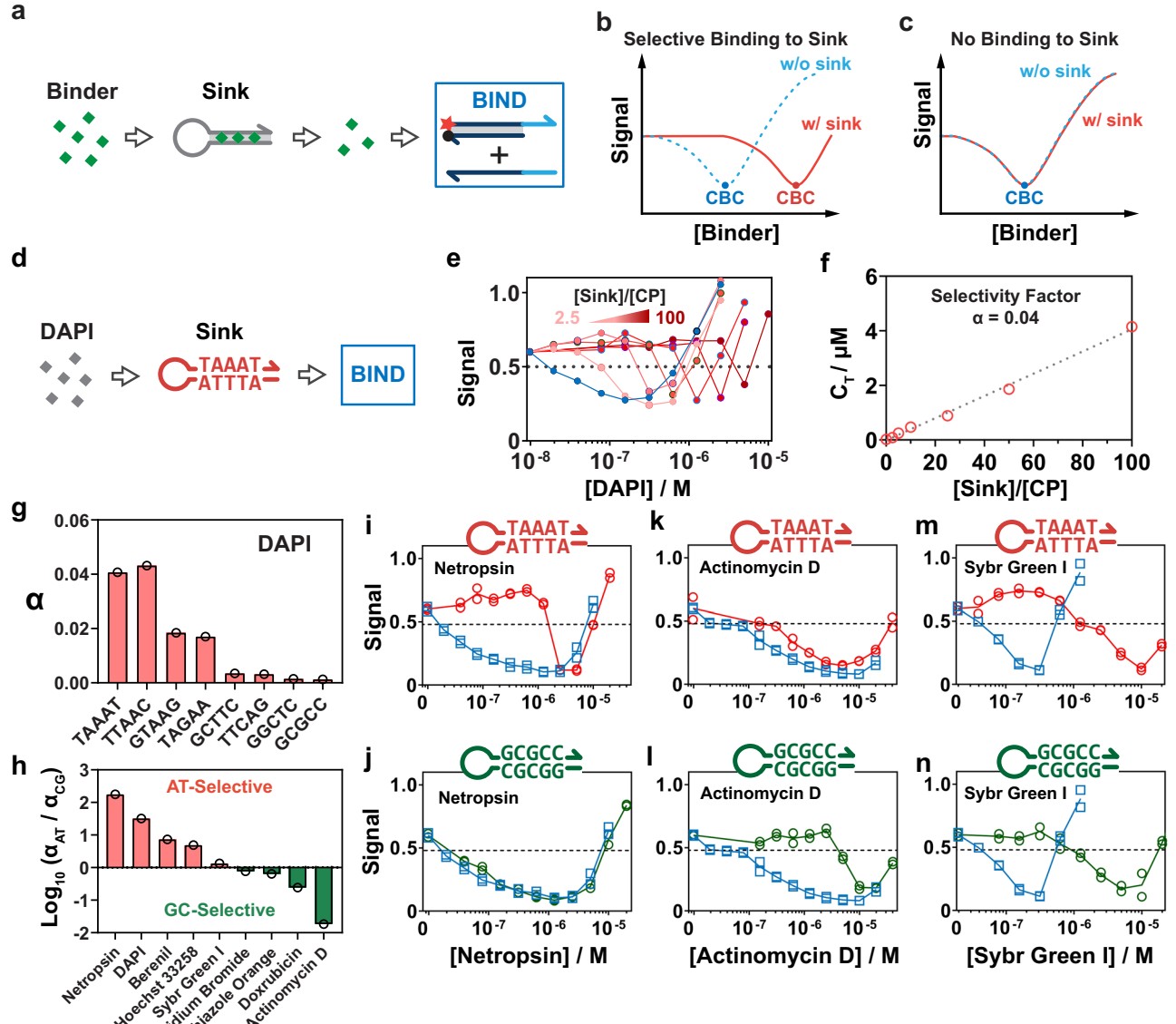

**Fig. 5 | Determining sequence selectivity of DNA-binder interactions using BIND.** **a** Schematic illustration of the competitive BIND reaction for determining the sequence selectivity of DNA binders. **b** Schematic illustration of the expected shift of the BIND curve for a binder with favored binding to the sink probe. **c** If the binder has no sequence selectivity to the sink probe, no shift is observed in the BIND curve. **d** Schematic illustration of determining the sequence selectivity of DAPI to a sink probe containing the 5′-TAAAT-3′ stem domain using BIND. **e** Experimentally observed shifts of BIND curves against increasing ratio of [sink]/ [CP] from 0 to 100. **f** Determining the selectivity factor of DAPI to the sink probe by plotting the critical concentration ($C_T$) of each BIND curve against the concentration ratio between sink and CP. **g** Quantitative profiling of the sequence selectivity of DAPI against sinks of increasing levels of GC content using BIND. **h** Quantitative profiling of the sequence selectivity of nine representative DNA binders against AT- and GC-rich sequences using BIND. Profiling the shifts in BIND curves against AT- and GC-rich sinks for netropsin (**i**, **j**), actinomycin D (**k**, **l**), and SYBR Green I (**m**, **n**).

designated 5 bp sequence and BIND probes. The selective binding of binders to the sink will reduce the effective concentration of binders to trigger the BIND reaction and thus shift the BIND curve as well as CBC toward a higher concentration range (Fig. 5b). On the other hand, no shift will be observed if there is no favored binding to the sink (Fig. 5c).

We first verified the effectiveness of BIND for evaluating the sequence selectivity using 4′,6′-diamidine-2-phenylindole (DAPI), which binds selectively to AT-rich sites in the minor groove (Fig. 5d)[31,32]. BIND curves of DAPI were found to keep shifting to higher concentration ranges when increasing the ratio between a 5′-TAAAT-3′ sink and BIND (probe CP) from 0 to 100 (Fig. 5e). We then defined the concentration of DAPI at 50% yield of displacement at the $S_N1$ domain as a threshold concentration ($C_T$). A linear relationship was then established when plotting $C_T$ against the ratios between sink and BIND

probes (Fig. 5f). We defined the slope as a selectivity factor α, which is used to quantify the degree of sequence selectivity. We measured and ranked the α of DAPI against 8 sink probes containing representative combinations of base pairs. The observed preferences of DAPI for a five ≈ four > three > two bp AT binding site in sink probes were consistent with previous studies using NMR[31] and fluorescent intercalator displacement (FID) assays (Fig. 5g and Supplementary Fig. 30)[32].

Having confirmed the effectiveness of BIND to evaluate the sequence selectivity using DAPI, we surveyed the sequence selectivity of eight other binders using a pair of AT-rich (5′-TAAAT-3′) and GC-rich (5′-GCGCC-3′) sink probes (Supplementary Figs. 31–38). By calculating the ratio of the selectivity factors of the two probes ($α_{AT}$/$α_{GC}$), we were able to determine and rank the sequence selectivity of all nine binders (Fig. 5h). Four binders were found to be selective for

AT-rich sequences ($\alpha_{AT}/\alpha_{GC} > 1$) with the rank order netropsin > DAPI > berenil > Hoechst 33258. Another four binders were found to be selective for GC-rich sequences ($\alpha_{AT}/\alpha_{GC} < 1$) with the rank order actinomycin D > doxorubicin > thiazole orange > ethidium bromide. Netropsin was found to be highly selective to the AT-rich sink and showed no binding to the GC-rich sequence even when the sink was 100 times in excess (Fig. 5i, j), whereas actinomycin D demonstrated strong selectivity to the GC-rich sequence and showed no binding to the AT-rich sequence (Fig. 5k, l). The sequence preference of actinomycin D was further confirmed by altering the GC content of the CP duplex, where highest affinity was determined for the duplex with highest GC content (75%) (Supplementary Fig. 39). BIND also revealed no sequence preference of SG-I for either AT- or GC-rich sequences with $\alpha_{AT}/\alpha_{GC}$ close to 1 (Fig. 5h, m, n).

## High-throughput screening of small-molecule DNA binders using BIND

Having demonstrated the comprehensive characterization of existing DNA binders using BIND, we aim to further engineer BIND as an HTS platform for discovering new DNA binders. Because BIND features a hyperbolic function, the same detection signal may correspond to a negative screening signal but may also correspond to a positive screening result (Supplementary Fig. 40). To avoid this confusion, we designed a tandem BIND assay in which $S_N1$ and $S_N2$ reaction pathways were programmed by adding two binders in tandem (Fig. 6a). Specifically, BIND was first induced by SG-I at a final concentration equal to its CBC so that minimal strand displacement occurred between CP and I (Fig. 6a, b). A secondary binder was then added to promote strand displacement via the $S_N2$ reaction pathway (Fig. 6a, b). The possibility of programming strand displacement reaction pathways using two tandem binders was successfully demonstrated using both SG-I/netropsin and SG-I/Ru(Phen)$_3$Cl$_2$ (Fig. 6c). More importantly, fluorescence signals were found to be specific to DNA binders and increased monotonically as a function of both binder affinity and concentration at the $S_N2$ domain, confirming that the tandem BIND assay is an ideal HTS platform for discovering new binders.

We then employed the tandem BIND assay for HTS of DNA binders. Netropsin and 1-BQC were used as the positive control (P. C.) and negative control (N.C.), respectively, throughout the screening. Using these two controls, each tested 10 times, a Z′ factor of 0.64 was determined, suggesting that the tandem BIND was of excellent assay quality for HTS (Fig. 6e). This was further confirmed upon validation against 15 existing DNA binders, where 12 hits and 3 misses were determined by BIND (Supplementary Fig. 41). The three misses included Hoechst 33258 due to the strong inner filter effect at the assay concentration (30 μM) and two neutral binders (actinomycin D and echinomycin) that could not promote $S_N2$ reaction pathways[33]. For comparison, the Z′ factor was determined to be only 0.39 for the classic FID screening assay using EB as an indicator (Supplementary Fig. 41). As a result of the narrow screening window or severe spectral overlap, only 6 out of 15 existing binders were determined to be positive hits using the FID screening assay (Supplementary Fig. 41).

We finally employed the tandem BIND assay to discover new binders by screening against a pool containing 700 compounds collected from Selleck's express pick library L3600 (Fig. 6d). This sublibrary is part of a collection provided by Pfizer of 4208 chemical compounds featuring different pathway inhibitors with high structural diversities. Screening these pathway inhibitors to identify DNA binders will expand our knowledge of their mechanism of action and the discovery of possible drug leads with antineoplastic and/or anti-infectious activities. A wide screening window was achieved between the sample and P. C. with a Z-factor of 0.54 (Fig. 6f). The close numerical values between the Z and Z′ factors suggest a negligible effect of the compound library on the tandem BIND assay[34]. A threshold of 0.25 ($\mu_S + 3\,\sigma_S$) was set, and 8 hits were successfully

returned (Supplementary Fig. 42) with the top four hits listed in Fig. 6g–j (S20 > J15 > H27 > A33). Of the 8 hits, S20 (2-(2-(diethylamino)ethyl)indeno[1,2,3-de]phthalazine-3(2H)-one) demonstrated the strongest activity in the tandem BIND assay. Comprehensive profiling of this compound using BIND returned a $K_d$ of 212 nM and a binding site size of 1.3 (Fig. 6k). The strong binding between S20 and dsDNA was further confirmed using classic melting analysis (Fig. 6l). We also determined the sequence selectivity of S20 using the competitive BIND assay, which shows no sequence preference with an $\alpha_{AT}/\alpha_{GC}$ value close to 1 (Supplementary Fig. 43). Thermodynamic analysis of S20 using BIND revealed a strong enthalpic contribution with ΔH° of −9.72 kcal/mol and a weak entropic contribution with TΔS° of −0.22 kcal/mol (Fig. 6m and Supplementary Fig. 43). The high enthalpic contribution suggests that S20 is likely to be an intercalator, which was echoed by molecular docking with a dsDNA model extracted from PDB file 108D (Fig. 6m). A cytotoxicity study revealed that S20 had broad-spectrum growth inhibitory activities against cancer cells, with inhibitory concentrations (IC 50) of 13.07 μM, 18.26 μM and 94.68 μM for HeLa, HepG2 and A549 cells, respectively (Supplementary Fig. 45).

## Discussion

In this work, we have demonstrated how DNA-binder interactions can be used to program reaction pathways of toehold-mediated DNA strand displacement reactions. Unlike previous applications using static binding of small molecules to dsDNA or DNA nanostructures, we achieved the direct and quantitative control of dynamic DNA nanotechnology by simply altering the type and/or concentration of DNA binders. Our success in developing BIND also provides mechanistic insights into DNA strand displacement reactions. Unlike existing strand displacement reactions that are dominated by the $S_N2$ reaction pathway, BIND possesses switchable reaction pathways from $S_N1$ to $S_N2$ in response to small-molecule DNA binders in a concentration-dependent manner. Distinct BIND profiles have also been established in accordance with binder affinities and charge conditions, which forms the fundamental basis for controlling dynamic DNA reactions using small-molecule binders.

BIND also provides a molecular platform for the comprehensive thermodynamic characterization of interactions between DNA and small molecules. Small molecules capable of binding to DNA through either intercalation or minor groove binding have long been an intensive focus of research because of their critical roles in therapeutics and biochemical research. Binding affinity, binding site size, sequence selectivity, and enthalpy and entropy changes are critical thermodynamic determinants of the binding behavior and functionality of DNA binders but remain difficult to measure on a single analytical platform. Instrumental methods, such as single-molecule force spectroscopy (SMFS)[35], differential scanning calorimetry (DSC)[36], isothermal titration calorimetry (ITC)[37], and thermal melting analysis[38], allow the accurate characterization of binding affinities but can seldom provide information on sequence selectivity. Moreover, these techniques often require specialized equipment and procedures that are not widely available in many biochemical laboratories. Although FID is a simple assay that allows the determination of sequence selectivity of DNA binders, it relies heavily on the relative binding strength between a given binder and an indicator (e.g., EB) and thus may introduce significant measurement bias upon usage[32,38]. BIND enables a simple, unbiased assay that allows the comprehensive characterization of all critical thermodynamic properties of DNA binders on a single platform. Using 16 well-studied DNA binders, we demonstrated that BIND not only led to highly consistent numeric values of the dissociation constant, binding site size, and enthalpy contribution of each binder with previous studies using varied characterization techniques but also offers quantitative information on the binding cooperativity and sequence selectivity through unique BIND profiles.

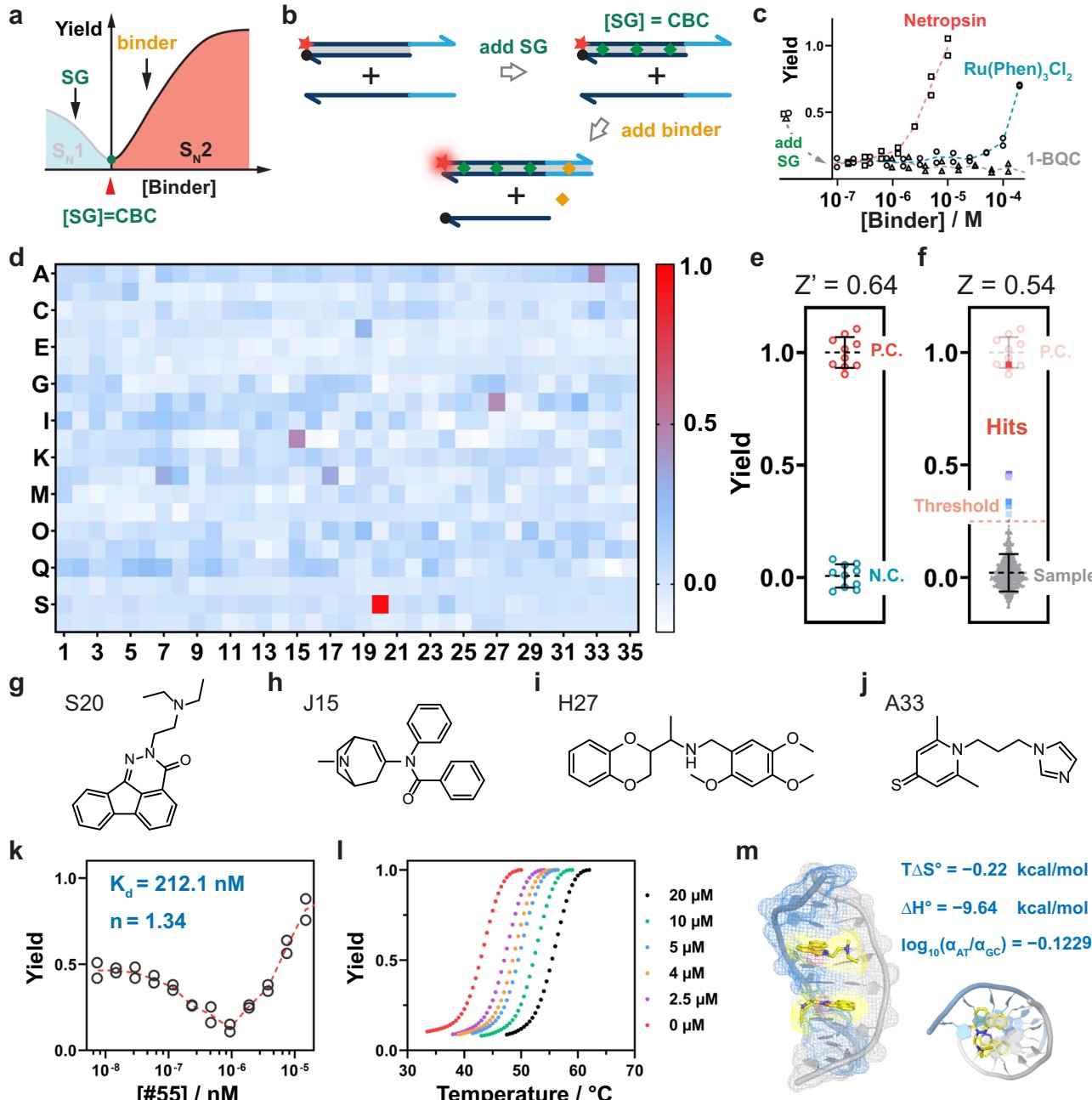

**Fig. 6 | High-throughput screening of new DNA binders using BIND. a** Design principle of the tandem BIND reaction as an HTS assay for new DNA binders. **b** Schematic illustration of the workflow for the tandem BIND. **c** Validation of the tandem BIND reaction against a representative strong binder (netropsin), weak binder (Ru(Phen)₃Cl₂), and nonbinder (1-BQC). **d** Results of the HTS BIND assay for a pool of 700 compounds. **e** Determining the Z′ factor of the tandem BIND assay using netropsin as a positive control and 1-BQC as a negative control. To determine Z′ value, technical replicates (*n* = 10) were used for both positive and negative controls. Each error bar represents one standard deviation from the 10 replicates. **f** Determining the Z factor that reflects the influence of the sample on the HTS. The error bar represents one standard deviation from the tests of all independent compounds (*n* = 700). **g**–**j** A list of the top four positive hits obtained from the tandem BIND assay. **k** Determining the binding affinity and binding site size of S20 using BIND. **l** Melting curves for dsDNA in the presence of varying concentrations of S20. **m** Critical thermodynamic parameters of S20 measured using BIND and the predicted binding mode using molecular docking. The DNA 3D image was extracted from PDB file 108D.

BIND also provides a high-quality and low-bias HTS platform for discovering new DNA binders. Although DNA binders are an important source of anticancer and anti-infective drugs, there have been very limited HTS approaches for finding new binders. To date, FID is the only HTS assay for DNA binders but is subject to low screening efficiency and significant screening bias because it relies on competitive binding between the binder and the indicator[39]. In contrast, we demonstrated that both strong and weak binders could effectively induce strand displacement reactions *via* the $S_N2$ reaction pathway and could thus be effectively identified using the engineered tandem BIND assay. Our success in finding 8 new binders from a pool of 700 candidate compounds further confirmed the power of BIND as an HTS platform for DNA binders. Moreover, because BIND is practically simple to use with no need for specialized equipment or procedures, we anticipate that BIND will be widely adopted as a one-stop HTS and comprehensive

characterization platform to accelerate the discovery of new DNA binders and DNA binding drugs.

As BIND is also a powerful characterization platform, we were able to further establish a comprehensive thermodynamic profile of the strongest hit, S20, in terms of its binding affinity, sequence selectivity, and enthalpy/entropy contributions to the binding thermodynamics. S20 was patented in 2012 as an interferon-inducing compound[40]. Despite the low interferon induction activity, S20 showed relatively high cytotoxicity to MDCK cells ($IC_{50} = 6.6\ \mu M$) compared to its derivative. We also found that S20 is also a strong DNA binder, which may contribute to its high cytotoxicity.

To further expand the concept and strategies of BIND in DNA nanotechnology and drug discovery, ongoing research focuses on two main challenges. First, the BIND system requires a low ionic strength below 25 mM and the absence of $Mg^{2+}$ (Supplementary Fig. 36), and this significantly limits its use in combination with existing strand displacement reactions, which are often performed in a standard buffer containing 12.5 mM $Mg^{2+}$. A possible solution to this challenge is to fine-tune the length of DNA probes so that BIND may work under standard buffer conditions. An alternative solution is to configure DNA reaction networks for buffer conditions compatible with BIND. Second, when BIND is used for the HTS of new binders, it excludes neutral binders since the promotion of the $S_N2$ reaction pathway requires both specific binding and positive charges. A solution to this challenge is to minimize screening bias by combining BIND and FID to develop an orthogonal HTS platform.

## Methods

### Binder-induced nucleic acid strand displacement

All DNA oligonucleotides were purchased from Integrated DNA Technologies (IDT, Coralville, IA, United States) and were purified by IDT using high-performance liquid chromatography. Sequences and modifications were listed in Supplementary Table 1. The BIND probe CP was prepared by heating a reaction mixture containing 5 μM probe C, 7.5 μM probe P, and 1 mM $Mg^{2+}$ in 1 × TE buffer at 95 °C for 5 min and then gradually cooling to room temperature at a constant rate over a period of 40 min using a Bio-Rad T100 thermocycler. The stock solution of CP at a final concentration of 5 μM was stored at 4 °C until use. For a typical BIND reaction, CP was diluted in 1× TE buffer and then mixed with a given DNA binder at 37 °C for 5 min. To this mixture, invader I was then added to initiate the BIND reaction at 37 °C. The fluorescence of the reaction mixture containing 10 nM I, 20 nM CP, and a given concentration of DNA binder was measured in real-time using a BioTek Cytation 5 Multimode Microplate reader at a data acquisition rate of one data point per minute for a period of 1 h. The excitation/emission wavelengths were set to 640 nm/675 nm. All fluorescence signals were normalized against a reaction mixture containing 20 nM CP, 10 nM I, and 10 mM $MgCl_2$ in 1 x TE buffer as a positive control. $Mg^{2+}$ (10 mM) was added to the positive control to ensure complete strand displacement. The solution containing 20 nM CP in 1 × TE buffer was also included as a negative control for fluorescence normalization. Data analyses were performed using Microsoft 365 Excel and GraphPad 8.0.1.

### Thermodynamic characterization of DNA-binder interactions using BIND

Endpoint fluorescence measurement was used to establish the BIND profile for measuring the binding affinities and binding site sizes of DNA-binder interactions. Briefly, CP was diluted in 1 × TE buffer and then mixed with a given DNA binder at 37 °C for 5 min. To this mixture, invader I was then added to initiate the BIND reaction. The reaction mixture containing 10 nM I, 20 nM CP, and a given concentration of DNA binder was incubated at 37 °C for 2 h before an endpoint fluorescence measurement using a BioTek Cytation 5 Multimode Microplate reader by setting the excitation/emission wavelengths to

640 nm/675 nm. The fluorescence signal was then normalized against the positive and negative controls as outlined above. The association binding constant Ka and binding site size n of each binder were then determined by fitting the fractional occupancy of bound binders using the McGhee and Von Hippel binding isotherm equation:

$$Y = \frac{FL_{sample} - FL_{background}}{FL_{P.C.} - FL_{background}} \tag{7}$$

$$O = 1 - \frac{Y_{sample} - Y_{CBC}}{Y_0 - Y_{CBC}} \tag{8}$$

$$O = \frac{n \cdot \left([binder]_{total} - [bp]_{CP} \cdot \frac{O}{n}\right) \cdot K_a \cdot (1 - O)^n}{\left(1 - O + \frac{O}{n}\right)^{(n-1)}} \tag{9}$$

where $F$ is the normalized fluorescence signal of a given sample, $F_{CBC}$ is the normalized fluorescence when the binder concentration equals its CBC, $F_0$ is the normalized fluorescence signal when no binder is added to the strand displacement system, O is the fractional occupancy of the bound binder, $C_{binder}$ is the total binder concentration at each sample, $C_{DNA}$ is the concentration of base pairs in CP (e.g., 20 nM CP consists of 320 nM base pairs), $K_a$ is the association constant and $n$ is the binding site size. When taking cooperativity $\omega$ into consideration, McGhee and Von Hippel's binding isotherm algorithm was transformed into:

$$O = K_a(1 - O) \cdot \left(\frac{2 \cdot \omega - 1) \cdot (1 - O) + \frac{O}{n} - L}{2 \cdot (\omega - 1) \cdot (1 - O)}\right)^{n-1} \cdot \left(\frac{1 - (n+1) \cdot \frac{O}{n} + L}{2 \cdot (1 - O)}\right)^2 \\ \cdot ([binder]_{total} - \frac{O \cdot [bp]_{CP}}{n}) \cdot n \tag{10}$$

where $L = \sqrt{\{[1 - (n+1)\frac{O}{n}]^2 + 4\omega\frac{O}{n}(1 - O)\}}$, and $\omega$ is the cooperativity.

Binding enthalpy and binding entropy were determined by measuring the Ka of a given biner using the BIND reaction at 27 °C, 30 °C, 32 °C, 35 °C, and 37 °C and then fit using Van't Hoff's equation:

$$\ln(K_a) = -\frac{\Delta H^\circ_{BIND}}{R} \cdot \frac{1}{T} + \frac{\Delta S^\circ_{BIND}}{R} \tag{11}$$

where $K_a$ is the association constant, $\Delta H$ is the binding enthalpy, $R$ is the universal gas constant, T is the temperature in Kelvin, and $\Delta S$ is the binding entropy.

### Determine the sequence selectivity of DNA binders using BIND

The sequence selectivity of DNA binders was determined using a competitive BIND reaction between CP and stem–looped sink probes. Each sink probe was designed to contain an 8 bp stem domain of varying ATGC combinations and a 5 nt poly-dA loop domain. For a typical competitive BIND reaction, a given DNA binder was premixed with a sink probe in 1 × TE buffer at 37 °C for 2 h. This reaction mixture was subsequently mixed with CP and I to initiate the BIND reaction using the protocol outlined above. The concentration ratios between the sink and CP probes were set to 2.5, 5, 10, 25, 50, and 100 in the competitive BIND reaction. The threshold concentrations ($C_T$) that were defined as the binder concentration at 50% displacement yield at the $S_N1$ domain were then plotted against the sink/CP ratios to determine the selectivity factor α, which was defined as the slope of the fitted linear curve.

### Kinetic characterization of $S_N1$ and $S_N2$ pathways in BIND

Kinetics of BIND was investigated using a strand-exchange approach developed by Reynaldo et al.[26] 20 nM CP duplex in the absence or presence of varying concentrations of SG-I was invaded using increasing concentration of I from 250 nM to 2 μM. The fluorescence of

the reaction mixture was measured in real-time using a BioTek Cytation 5 Multimode Microplate reader at a data acquisition rate of one data point per minute for a period of 1 h. The excitation/emission wavelengths were set to 640 nm/675 nm. Raw fluorescence of each data point was normalized against a positive control and a negative control as described above to determine the reaction yield. The observed rate constant was then calculated by fitting the normalized experimental data against the following equation: $[CP]_t = [CP]_0 \cdot \exp(-k_{obs} \cdot t)$. Rate constants $k_{dissociative}$ and $k_{displacement}$ were then determined through a linear regression against the theoretical model developed by Reynaldo et al.[26]:

$$k_{obs} = k_{dissociative} + k_{displacement} \cdot [I]$$

### High-throughput screening of small-molecule DNA binders using BIND

Tandem BIND reactions performed in parallel in 96-well microplates were used to establish the HTS assay for discovering new DNA binders from a pool of 700 compounds collected from Selleck's express pick library. Briefly, the fluorescence signal of each BIND reaction mixture containing 20 nM CP and 10 nM I in 1 × TE buffer was suppressed by adding SG-I at a final concentration equal to its CBC (250 nM) and incubated at 37 °C for 10 min. To this reaction mixture, a candidate compound at a final concentration of 30 μM was added. After another incubation at 37 °C for 2 h, endpoint fluorescence was measured using a BioTek Cytation 5 Multimode Microplate reader at excitation/emission wavelengths of 640 nm/675 nm. The fluorescence signal in each well was normalized using the positive and negative controls as outlined above. Compounds with numeric values above the threshold (0.25) were considered positive hits and were subjected to subsequent thermodynamic characterization using BIND and thermal melting analysis.

### Reporting summary

Further information on research design is available in the Nature Portfolio Reporting Summary linked to this article.

## Data availability

All fluorescence measurement data generated in this study are provided in Supplementary Information and Source Data file. Source data are provided with this paper.

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

## Acknowledgements

We thank the Fundamental Research Funds for the Central Universities (no. YJ201975, F.L.), the National Natural Science Foundation of China (22074099, F.L.), and the Institutional Research Fund from Sichuan University (2021SCUNL105, F.L.). We thank Professor Xinxin Feng and Ms. Junfeng Song for helping test the cellular toxicity of BIND-screened binders.

## Author contributions

F.L. conceived the idea and supervised the overall project. J.X. designed and performed all experiments to characterize BIND reactions. J.X. and L.W. performed all BIND assays for the comprehensive characterization of existing and newly discovered DNA binders. G.A.W., and L.G. performed all in silico experiments, including building the theoretical model, data fitting, simulation, and molecular docking. J.X., Q.L., and H.D. designed and performed the tandem BIND reactions for the high-throughput screening of new DNA binders from the chemical library. All authors participated in the drafting and revision of the manuscript.

## Competing interests

J.X. and F.L. are inventors of a pending patent (NO. 202310363284X). All other coauthors declare no competing interests.
