## [Peer review file · Nature Communications]

REVIEWER COMMENTS

Reviewer #1 (Remarks to the Author):

Xu et al. developed a novel method to regulate dynamic DNA reactions through DNA binders. They systematically studied the influence of DNA binders on strand displacement reaction. Based on the findings, they established a platform to screen new DNA binders. This work represents an advance in dynamic DNA nanotechnology. Revisions are suggested as following:

1. In the model proposed by the authors in Fig 1, SN1 pathway was activated in the absence of binder. However, it is also possible that the CI duplex was not formed and the final-state product consists of three single strands.
2. The fluorescence intensity of Cy5 was used to measure the yield of BIND with different binders, which avoided the effect of differences in fluorescence properties between binders on results. Has the authors tested the influence of binder concentration and type on the fluorescence of Cy5? Besides, some binders may have weak emission at 675 nm, would this affect the determination of the Cy5 signal?
3. The authors used 10 nM of I and 20 nM of CP in their tests. Why should CP be excessive? The CBC obtained was on the order of μM , while the concentration of the DNA strand was on the order of 10 nM. Could the authors interpret this concentration relationship between binders and DNA strands.
4. There are a number of formulas and corresponding descriptions. To increase readability, some non-central formulas and their reasoning process could be moved to Methods, and refer them in the maintext with easy-to-understand expressions.
5. The authors established a tandem BIND assay to screen new binders. It could be more convincing if the author could explain the necessity and advantages of adding two binders in tandem.

Minor issues:

1. In line 102, the authors wrote $k_{sn1} = k_d \times k_n$. Please classify how this was obtained.
2. The abbreviation "BIDN" in Abstract should be "BIND".

Reviewer #2 (Remarks to the Author):

Xu et al present a novel method called BIND(Binder-Induced Nucleic acid strand Displacement) to measure and optimize kinetics and thermodynamics of DNA-binder interactions utilizing the inhibition of DNA dissociation and enhancement of DNA strand displacement migration. They first test multiple well-known binder molecules to support BIND mechanism, and the BIND showed distinctive performance under variety of charge conditions and binding affinities. Deciphering with the lattice-ligand model, the authors are able to measure DNA-binder kinetics and thermodynamics, and highly concordant with previous studies using other equipments. The tune-ability and binder-sequence selectivity of BIND is demonstrated by addition of hairpin-structured sink oligos. At last, an application using BIND is showed to identify new potential binders. Given the fundamental importance of DNA interactions, and BINDs is capable of exploring new binders, this work is suitable for publication in Nature Communications. However, there are a number of general and more specific points that need to be addressed.

General comments:

1. There is only one set of CP + I used in this whole paper, and the authors already proved that binder-DNA interaction is related to binding size and sequences. It will be great if the authors can perform experiment using just one binder, but different sets of CP+I to answer

following questions:

a. How will the length of DNA probes(CP and I) impact on BIND model? Will there be blending-in between SN1 and SN2?

Generally speaking, DNA breathing is common for long CP complex, but hard to fully dissociate with metal cations. Will CBP changed if using a different length CP probe? Or is $CBP/DNA_probe_length = \text{constant}$?

b. Considering binders have selectivity over GC contents, will changing the C to more GC rich or AT-rich(same length) will influence on: 1) BIND curves, 2) K_d , 3) thermodynamics?

c. If there is no change in K_d , please include the reproducibility of BIND using different probes in the supplement.

If there is change in K_d , please include explanation in discussion.

2. It is helpful to include a little more explanation in fitting K_d and n . The equation(5), line 186, is not clearly related to binder concentrations and fluorescence yield. Please include detail calculation and fitting process in supplementary.

Also, in line 481, the formula used for Y calculation doesn't make sense.

When $F = 0$, in absence of binder, $Y = 0$, $C_{bind} = 0$, the equation is correct.

However, when $F = F_{CBC}$, so $Y = 1$, but on the right side of this equation is $(1-Y)^n = 0$, the equation doesn't balance.

Please correct and include calculations of transforming equation(5) to line 481.

3. Some questions regarding tandem BIND.

a. Why tandem BIND? As showed in Fig. S6, there will be reduction of yield in Sn1 stage if you directly introduce binder at 0.1uM. What is the benefit of using tandem BIND?

b. Cross-referencing Fig.2I and Fig. 6c, why the CBC of RuPhen3Cl2 is delayed by 2 order of magnitude? This actually show that screening binders using tandem BIND with fixed binder conc may not directly reflect binder CBC, or even miss some binders.

c. The application of BIND is based on saturating Sn1 stage, that will exclude all neutral binders considering they only work on Sn1. Would you please address how BIND can work on this?

4. Regarding the analysis of C_T vs $[Sink]/[CP]$, CBC shall be more obviously delayed with more addition of Sinks, is there any correlation between CBC vs $[Sink]/[CP]$?

4. In Fig.5e, the trace of $[Sink]/[CP] = 2.5$ showed a lower Yield at CBC. Generally speaking, I would assume the competition between Sink and CP will make binder-CP less efficient, resulting in a less yield or larger CBC. Is this result reproducible? If so, please explain if there is any mechanism with additional sink can strengthen binder-DNA interaction?

5. Why mention K_{sn1}/K_{sn2} when there is no discussion related. Better remove them for easier readability.

Specific comments:

1. Confusion of terms: K_D , K_d , K_a and K_{eq}

The authors started with explaining pure DNA dissociation reaction, where equilibrium constant is K_D . However, the whole paper is discussing K_d and K_a , and in my understanding it is representing the equilibrium constant of DNA-binder interactions, better add some upper suffix to better distinguish K_D and K_d . Also, K_a is $(1/K_d)$, so better consistently using one symbol across the whole paper instead of using K_a in the text and K_d in the figures.

2. Confusion of lines in figures.

In Fig.2 j-m, Fig.3 e-f and lots of supplementary figures, the authors used a line to link all experimental data points, it is confusing to use lines without explanation, is it median or avg? Or is is a simulated curve? Please keep this line format consistency.

Considering the author has figures with linear regression fitting lines in solid black, I recommend to use another line type.

3. Include $[Sink]$ and $[CP]$ table in the supplementary table.

3. Detailed comments:

1) typo at following lines:

Line 25: BIND

Line 179: an

Fig. 3a: the ylabel should be Yield? If plotting against occupancy, occupancy will be 0 when binder conc equals to CBC.

Fig. C/e/g, In Keq should be In K_a ?

Reviewer #3 (Remarks to the Author):

This is a very interesting study on modulating toehold-mediated DNA strand displacement with small-molecule DNA binders. The authors show that at very low salt conditions and with divalent ions sequestered, the efficiency of strand displacement reactions (measured by the yield of expected products) can be quantitatively controlled by DNA binders. They went on to characterize the thermodynamics of a panel of known DNA binders as well as screening for new DNA binders using this platform (termed BIND).

Discovery and characterization of DNA binders are important for drug development. Although existing methods are mature, this new platform is welcomed because it potentially offers both high-throughput screening and thermodynamic characterization capabilities on a wider range of DNA-binding molecules. Additionally, because strand displacement is a one of the key mechanisms in dynamic DNA nanotechnology, the BIND platform may be used to control the activities of DNA nanodevices, though this avenue was not explored in this study.

The claims in this manuscript were mostly supported by fluorescence dequenching data that were used to derive the yield of the strand displacement reaction. This is an well established methodology appropriately used for this study.

Nevertheless, I have some concerns about the manuscript, as follows:

(1) A major claim is that DNA-binder interactions can be used to program reaction pathways of DNA strand displacement. Although this is likely true, there are essentially no data that support this claim. In this study, all measurements are derived from the the end-point fluorescence, which can only inform on the thermodynamics (binding constant) but not the kinetics of the reactions. A good example is ref 26, where the authors analyzed the contributions of the two pathways using the kinetics data from time-course studies. The authors appeared to have acquired fluorescence traces over time (Fig. S1), but for some reason chose to only analyze the end point. As a result, no rate constant was determined, and unfortunately that does not support the claims regarding the reaction pathways (e.g. Sn1 vs Sn2) and their kinetics (e.g. "acceleration").

(2) The way the Results section is written can lead to confusion because of insufficient experimental details described in this section. I suggest the authors clarify at the beginning of this section how the assay was set up before presenting data and discussing the results. Details need to be covered include the sequence, concentration, order/timing of addition of the DNA species and the exact buffer condition. I am also puzzled by the 1 M EDTA in the 1x TE buffer (in Supplementary Experimental Section).

(3) The authors argues that charge is one of the two determinants of whether a DNA-binder promotes the strand displacement after saturating the pre-formed DNA duplex. This explanation is not quite satisfying. One has to ask why charge is important for promoting

strand displacement on a binder-stabilized dsDNA? The neutral molecules have $K_d > 100$ nM. Calling them "strong binder", a name also used for molecules with $K_d < 10$ nM, is hard to justify. Perhaps this is another place where the authors may want to examine the kinetics of the strand displacement reactions.

(4) Out of the 20 small molecules, 4 of them did not return measurable K_d or n (Fig. S6). I commend the authors for including these data, but at the same time, urge them to discuss these phenomena.

Minor points:

(5) The term "sequence preference" and "sequence selectivity" were used many times, leaving the impression that the DNA binders were systematically evaluated against a comprehensive library of DNA sequences. However, only two 5-bp long DNA sequences were tested. So at best, the data reflect the DNA binder's selectivity for GC-rich sequences. This needs to be clarified throughout the manuscript.

(6) What are the color codes used for the DNA binder names in Fig. 3. Why are there 4 groups?

(7) Line 145, Fig. 4j and Fig. 4k should probably be Fig. 2j and Fig. 2k.

Response Letter

We thank all reviewers for their insightful comments and suggestions. We have thoroughly revised our manuscript accordingly. A point-by-point response is provided as follows:

Reviewer #1

Xu et al. developed a novel method to regulate dynamic DNA reactions through DNA binders. They systematically studied the influence of DNA binders on strand displacement reaction. Based on the findings, they established a platform to screen new DNA binders. This work represents an advance in dynamic DNA nanotechnology. Revisions are suggested as following:

Comment-1. In the model proposed by the authors in Fig 1, SN1 pathway was activated in the absence of binder. However, it is also possible that the CI duplex was not formed, and the final-state product consists of three single strands.

Response-1: For the initial CP duplex, we found that ~10% were dissociated based on the fluorescence signal in the absence of the invader (Supplementary Fig. S2h). As CI duplex contained 7 more base pairs than the CP duplex, we believe that the CI duplex was formed with even smaller portion dissociated into C and I.

Comment-2. The fluorescence intensity of Cy5 was used to measure the yield of BIND with different binders, which avoided the effect of differences in fluorescence properties between binders on results. Has the authors tested the influence of binder concentration and type on the fluorescence of Cy5? Besides, some binders may have weak emission at 675 nm, would this affect the determination of the Cy5 signal?

Response-2: We thank the reviewer for this concern and agree that it is critical to evaluate potential influence of binder concentration and type on the fluorescence of Cy5. In the revised manuscript, we added the measured fluorescence of 10 nM Cy5-labeled CI duplex in the presence of high concentrations (5 μ M) of varying types of binders. All fluorescence signals were normalized against Cy5-labeled CI duplex in the absence of DNA binders. The results suggest that there was no interference to the Cy5 signal at the emission of 675 nm.

Revision-2: Supplementary Figure S9 was added in the revised manuscript.

Fig. S10 | Evaluation of the potential influence of binder on fluorescence of Cy5. The Cy5-labeled CI duplex was fixed at 10 nM and binder concentrations were fixed at 5 μ M. All fluorescence signals were normalized against 10 nM Cy5-labeled CI duplex in the absence of DNA binders.

Comment-3. The authors used 10 nM of I and 20 nM of CP in their tests. Why should CP be excessive? The CBC obtained was on the order of μ M, while the concentration of the DNA strand was on the order of 10 nM. Could the authors interpret this concentration relationship between binders and DNA strands.

Response-3: In principle, DNA strand displacement reaction works for any reaction stoichiometry. Here, we intentionally keep CP in excess, so that the invader I is the limiting reagent. We keep the stoichiometry between CP and I to be 2:1, so that it is easier to establish and evaluate the mathematical model. Because the K_d values of most binders were at the μ M range, we chose a low nM DNA duplex concentration to ensure sufficient occupancy of binders at high binder concentrations.

Comment-4. There are a number of formulas and corresponding descriptions. To increase readability, some non-central formulas and their reasoning process could be moved to Methods, and refer them in the main text with easy-to-understand expressions.

Response-4: We thank the reviewer for this suggestion. We moved all non-central formulas, including McGhee and Von Hippel binding isotherm and Van't Hoff's fitting to the Methods. We also made

changes of expressions and symbols to increase readability.

Comment-5. The authors established a tandem BIND assay to screen new binders. It could be more convincing if the author could explain the necessity and advantages of adding two binders in tandem.

Response-5: It is necessary to use a tandem BIND assay because BIND features a hyperbolic curve. If BIND was directly used to perform high throughput screening, the same detection signal may correspond to a negative screening result at the S_{N1} domain but may also correspond to a positive result at the S_{N2} domain (a). To avoid this confusion, the tandem BIND was designed, so that there was a low fluorescence signal at the CBC of the known binder (b). Upon screening, any fluorescence increase corresponds to the binding of the second binder and returns a positive screening result. We added the cartoon in the revised manuscript as Supplementary Fig. S40 to better illustrate the necessity of tandem BIND.

Revision-5: Supplementary Fig. S40 and corresponding discussion were added in the revised manuscript.

Fig. S40 | Schematic illustration of the necessity of tandem BIND. It is necessary to use a tandem BIND assay because BIND features a hyperbolic curve. If BIND was directly used to perform high throughput screening, the same detection signal may correspond to a negative screening result at the S_{N1} domain but may also correspond to a positive result at the S_{N2} domain (a). To avoid this confusion, the tandem BIND was designed, so that there was a low fluorescence signal at the CBC of the known binder. Upon screening, any fluorescence increase corresponds to the binding of the second binder and returns a positive screening result.

Minor issues:

Comment-6. In line 102, the authors wrote $k_{sn1} = k_D \times k_H$. Please classify how this was obtained.

Response-6: Because the overall, the net reaction is:

where K_{sn1} is the equilibrium constant of this net reaction formula.

The same net reaction can also be written as two elementary reactions:

where K_D and K_H were the equilibrium constants for each reaction. According to the fundamentals of chemical equilibrium. $K_{sn1} = K_D \times K_H$

Comment-7. The abbreviation "BIDN" in Abstract should be "BIND".

Response-7: Corrected as suggested.

Reviewer #2

Xu et al present a novel method called BIND(Binder-Induced Nucleic acid strand Displacement) to measure and optimize kinetics and thermodynamics of DNA-binder interactions utilizing the inhibition of DNA dissociation and enhancement of DNA strand displacement migration. They first test multiple well-known binder molecules to support BIND mechanism, and the BIND showed distinctive performance under variety of charge conditions and binding affinities. Deciphering with the lattice-ligand model, the authors are able to measure DNA-binder kinetics and thermodynamics, and highly concordant with previous studies using other equipments. The tune-ability and binder-sequence selectivity of BIND is demonstrated by addition of hairpin-structured sink oligos. At last, an application using BIND is showed to identify new potential binders. Given the fundamental importance of DNA interactions, and BINDs is capable of exploring new binders, this work is suitable for publication in Nature Communications. However, there are a number of general and more specific points that need to be addressed.

General comments:

Comment-1: There is only one set of CP + I used in this whole paper, and the authors already proved that binder-DNA interaction is related to binding size and sequences. It will be great if the authors can

perform experiment using just one binder, but different sets of CP+I to answer following questions:

Comment-1a: How will the length of DNA probes (CP and I) impact on BIND model? Will there be blending-in between SN1 and SN2? Generally speaking, DNA breathing is common for long CP complex, but hard to fully dissociate with metal cations. Will CBC changed if using a different length CP probe? Or is $CBC/DNA_probe_length = constant$?

Response-1a: We thank the reviewer for raising this interesting question. In the revised manuscript, we added an investigation of the impact of DNA duplex length on BIND. Specifically, we established BIND profiles for three duplex lengths (all with the same level of GC content), 11 bp, 16 bp, and 21 bp. We found that the initial fluorescence in the absence of binder decreased as the length of duplex increased. This is to be expected as more complementary base pairs made CP duplex more stable even in the absence of metal cations. We also found that the CBC shifted from a lower concentration of 168 nM to 328 nM when increasing the duplex length from 11 bp to 21 bp. Meanwhile, the ratio between CBC and duplex length was a constant. In the revised manuscript, we added this result as Supplementary Fig. S3 in the revised manuscript.

Revision-1a: Supplementary Fig. S3 and corresponding discussion were added in the revised manuscript.

Fig. S3 | Effect of duplex length on BIND. BIND profiles were established for three duplex lengths (all with the same level of GC content), including 11 bp, 16 bp, and 21 bp. The initial fluorescence in the absence of binder was found to decrease as the length of duplex increased. This is to be expected as more complementary base pairs made CP duplex more stable even in the absence of metal cations. CBC was found to shift from a lower concentration of 168 nM to 328 nM when increasing the duplex length from 11 bp to 21 bp. Meanwhile, the ratio between CBC and duplex length was a constant.

Comment-1b: Considering binders have selectivity over GC contents, will changing the C to more GC rich or AT-rich(same length) will influence on: 1) BIND curves, 2) K_d , 3) thermodynamics?

Response-1b: We also investigated the impact of GC content on BIND and added these results in the revised manuscript. For a GC-selective binder Actinomycin D, the initial fluorescence in the absence of binder decreased as the GC content of duplex increased. This is also to be expected as the CP duplex becomes more stable with higher GC content. Both CBC and K_d values shifted to lower concentrations as we increased the GC content of the duplex. This observation was consistent with the high GC selectivity of Actinomycin D. Thermodynamically, changing the GC content exerts the impact on the binding enthalpy which changed from 3.44 kcal/mol at 25% GC content to 0 kcal/mol at 75% GC content.

Revision-1b: Supplementary Fig. S39 was added in the revised manuscript.

Fig. S39 | Impact of GC content on BIND. For a GC-selective binder Actinomycin D, the initial fluorescence in the absence of binder decreased as the GC content of duplex increased. This is also to be expected as the CP duplex becomes more stable with higher GC content. Both CBC and K_d values shifted to lower concentrations as we increased the GC content of the duplex. This observation was consistent with the high GC selectivity of Actinomycin D. Thermodynamically, changing the GC content exerts the impact on the binding enthalpy which changed from 3.44 kcal/mol at 25% GC content to 0 kcal/mol at 75% GC content.

Comment 1c: If there is no change in K_d , please include the reproducibility of BIND using different probes in the supplement. If there is change in K_d , please include explanation in discussion.

Response-1c: There is a significant change in K_d for DNA duplexes with varying GC content for Actinomycin D. This is because this binder binds preferentially to GC-rich DNA duplex and has a higher binding affinity for a duplex with high GC content.

Revision-1c: We added the corresponding discussion in the revised manuscript.

Comment-2: It is helpful to include a little more explanation in fitting K_d and n . The equation(S), line 186, is not clearly related to binder concentrations and fluorescence yield. Please include detail calculation and fitting process in supplementary. Also, in line 481, the formula used for Y calculation doesn't make sense. When $F = 0$, in absence of binder, $Y = 0$, $C_{bind} = 0$, the equation is correct. However, when $F = F_{CBC}$, so $Y = 1$, but on the right side of this equation is $(1-Y)^n = 0$, the equation doesn't balance. Please correct and include calculations of transforming equation(S) to line 481.

Response-2: To better illustrate the process of data fitting, we added Figures S11 and S14 in the Supplementary Material of the revised manuscript.

Supplementary Fig. S11 illustrates the process for determining K_d and n using the S_{N1} domain of BIND curve. In a typical workflow, BIND at the S_{N1} domain was first extracted and the reaction yield was converted into Occupancy (O) using the equation shown in Fig. S11. Occupancies at the range of 0.3 to 0.8 were then used for data fitting because of the limitation of McGhee and Von Hippel binding isotherm. K_d and n could then be determined through the fitting process.

For determining the thermodynamic parameters, the abovementioned fitting process was performed for BIND at varying temperatures. The K_d values were then converted into K_a which were further fit using Van't Hoff equation to determine ΔH and ΔS .

We also agree with the reviewer that McGhee and Von Hippel binding isothermal does not work when the occupancy is very high or very low. This is the reason why we chose an occupancy range between 0.3 and 0.8 to perform the fitting.

Revision-2: Supplementary Figures S11 and S14 were added into the revised manuscript to detail the

process of data fitting for determining K_d , n and key thermodynamic parameters.

Fig. S11 | Schematic illustration the workflow and mathematical transformation to determine the K_d and n of varying binders using BIND. The fractional occupancy from 30% and 80% was determined to be the optimal range for determining critical binding parameters, including dissociation constants, and binding site sizes.

Fig. S14 | Schematic illustration the workflow of data fitting to key thermodynamic parameters.

Comment-3: Some questions regarding tandem BIND.

Comment-3a: Why tandem BIND? As showed in Fig. 6, there will be reduction of yield in S_{N1} stage if you directly introduce binder at 0.1 μM . What is the benefit of using tandem BIND?

Response-3a: It is necessary to use a tandem BIND assay because BIND features a hyperbolic curve. If BIND was directly used to perform high throughput screening, the same detection signal may correspond to a negative screening result at the S_{N1} domain but may also correspond to a positive result at the S_{N2} domain (a). To avoid this confusion, the tandem BIND was designed, so that there was a low fluorescence signal at the CBC of the known binder (b). Upon screening, any fluorescence increase corresponds to the binding of the second binder and returns a positive screening result. We added the cartoon in the revised manuscript as Supplementary Fig. S40 to better illustrate the necessity of tandem BIND.

Revision-3a: Supplementary Fig. S40 was added in the revised manuscript.

Fig. S40 | Schematic illustration of the necessity of tandem BIND. It is necessary to use a tandem BIND assay because BIND features a hyperbolic curve. If BIND was directly used to perform high throughput screening, the same detection signal may correspond to a negative screening result at the S_{N1} domain but may also correspond to a positive result at the S_{N2} domain (a). To avoid this confusion, the tandem BIND was designed, so that there was a low fluorescence signal at the CBC of the known binder. Upon screening, any fluorescence increase corresponds to the binding of the second binder and returns a positive screening result.

Comment-3b: Cross-referencing Fig.2I and Fig. 6c, why the CBC of RuPhen3Cl2 is delayed by 2 order of magnitude? This actually show that screening binders using tandem BIND with fixed binder

cone may not directly reflect binder CBC, or even miss some binders.

Response-3b: We thank the reviewer for pointing out this inconsistency. We double checked our source data and found that the scale of Fig. 2I was incorrect. RuPhen₃Cl₂ is a very weak binder, so the correct CBC was at 25 μ M which was consistent with the result in Fig. 6c.

Revision-3b: Fig. 2I was replaced with the correct result in the revised manuscript.

Comment-3c: The application of BIND is based on saturating Sn1 stage, that will exclude all neutral binders considering they only work on Sn1. Would you please address how BIND can work on this?

Response-3c: The reviewer is correct that the tandem BIND assay excluded all neutral binders during the high-throughput screening, and we stated this limitation by the very end of the manuscript. We are currently working on new strategies that can screen both neutral and charged binders and this is out of the scope of the present manuscript.

Comment-4: Regarding the analysis of C_T vs [Sink]/[CP], CBC shall be more obviously delayed with more addition of Sinks, is there any correlation between CBC vs [Sink]/[CP]?

Response-4: We agree with the reviewer that in principle CBC should be more obviously delayed with the addition of sink probes. Practically, however, it was very difficult for us to accurately find CBC in all cases. That is the reason why we chose a threshold value rather than CBC to determine the selectivity factor. So, there is a correlation between CBC and [Sink]/[CP] (as shown in the Figure below), which shows the same trend as the threshold concentration v.s. [Sink]/[CP].

Comment-5: In Fig.5e, the trace of $[Sink]/[CP] = 2.5$ showed a lower Yield at CBC. Generally speaking, I would assume the competition between Sink and CP will make binder-CP less efficient, resulting in a less yield or larger CBC. Is this result reproducible? If so, please explain if there is any mechanism with additional sink can strengthen binder-DNA interaction?

Response-5: We thank the reviewer for raising this concern. We found that the issue was caused by the first data point (in the absence of the binder) for this set of experiment ($[Sink]/[CP] = 2.5$), which was abnormally higher. So, during the data normalization, values of all data points were lower than their actual values (e.g., baseline was lower than all other sets of experiments). In the revised manuscript, we corrected this issue by repeating and replacing the data set for $[Sink]/[CP] = 2.5$. All $[Sink]/[CP]$ sets show similar yields at CBC.

Revision-5: Fig. 5e was replaced with the newer version in the revised manuscript.

Comment-6: Why mention K_{sn1}/K_{sn2} when there is no discussion related. Better remove them for easier readability.

Response-6: We remove discussion of K_{sn1} and K_{sn2} from the main content. These parameters were mainly involved in building the mathematical models and were detailed in the Supplementary Materials.

Specific comments:

Comment-7: Confusion of terms: K_D , K_d , K_a and K_{eq} . The authors started with explaining pure DNA dissociation reaction, where equilibrium constant is K_D . However, the whole paper is discussing K_d and K_a , and in my understanding, it is representing the equilibrium constant of DNA-binder interactions, better add some upper suffix to better distinguish K_D and K_d . Also, K_a is $(1/K_d)$, so better consistently using one symbol across the whole paper instead of using K_a in the text and K_d in the figures.

Response-7: We thank the reviewer for this suggestion. We have thoroughly revised manuscript to keep using K_d to describe the affinity of DNA binders throughout the manuscript. K_a was only used when Van't Hoff equation was used to determine key thermodynamic parameters.

Comment-8: Confusion of lines in figures. In Fig.2 j-m, Fig.3 e-f and lots of supplementary figures, the authors used a line to link all experimental data points, it is confusing to use lines without explanation, is it median or avg? Or is it a simulated curve? Please keep this line format consistency. Considering the author has figures with linear regression fitting lines in solid black, I recommend to use another line type.

Response-8: We thank the reviewer for the suggestion. These lines were just to connect the mean of scatter the data points to build a BIND curve. To be different from linear fitting in solid black, we used red dashed lines to connect the means in Fig. 2 j-2m and Fig. 3e-3f.

Comment-9: Include [Sink] and [CP] table in the supplementary table.

Response-9: The sink and CP sequences were included as Supplementary Table 1.

Comment-10: Detailed comments: typo at following lines: Line 25: BIND; Line 179: an. Fig. 3a: the ylabel should be Yield? If plotting against occupancy, occupancy will be 0 when binder conc equals to CBC. Fig. 4c/e/g, $\ln K_{eq}$ should be $\ln K_a$?

Response-10: Corrected as suggested by the reviewer.

Reviewer #3

This is a very interesting study on modulating toehold-mediated DNA strand displacement with small-molecule DNA binders. The authors show that at very low salt conditions and with divalent ions sequestered, the efficiency of strand displacement reactions (measured by the yield of expected products) can be quantitatively controlled by DNA binders. They went on to characterize the thermodynamics of a panel of known DNA binders as well as screening for new DNA binders using this platform (termed BIND).

Discovery and characterization of DNA binders are important for drug development. Although existing methods are mature, this new platform is welcomed because it potentially offers both high-throughput screening and thermodynamic characterization capabilities on a wider range of DNA-binding molecules. Additionally, because strand displacement is a one of the key mechanisms in dynamic DNA

nanotechnology, the BIND platform may be used to control the activities of DNA nanodevices, though this avenue was not explored in this study.

The claims in this manuscript were mostly supported by fluorescence dequenching data that were used to derive the yield of the strand displacement reaction. This is an well established methodology appropriately used for this study.

Nevertheless, I have some concerns about the manuscript, as follows:

Comment-1: A major claim is that DNA-binder interactions can be used to program reaction pathways of DNA strand displacement. Although this is likely true, there are essentially no data that support this claim. In this study, all measurements are derived from the end-point fluorescence, which can only inform on the thermodynamics (binding constant) but not the kinetics of the reactions. A good example is ref 26, where the authors analyzed the contributions of the two pathways using the kinetics data from time-course studies. The authors appeared to have acquired fluorescence traces over time (Fig. S1), but for some reason chose to only analyze the end point. As a result, no rate constant was determined, and unfortunately that does not support the claims regarding the reaction pathways (e.g. Sn1 vs Sn2) and their kinetics (e.g. "acceleration").

Response-1: We thank the reviewer for raising this critical concern. Indeed, reaction pathways could be better characterized using kinetic analysis. In the revised manuscript, we systematically investigated the kinetic performance of BIND as suggested by the reviewer. We summarized these results as Supplementary Figures S4-S7. We confirmed that kinetic profiles of BIND fully agreed with our proposed reaction pathways.

Revision-1: Supplementary Figures S4-S7 and corresponding discussions were added in the revised manuscript.

Fig. S4 | Schematic illustration of the workflow for investigating the kinetics of BIND. Kinetic curves were collected when invading CP duplex using increasing concentrations of invader I. The observed rate constant k_{obs} were determined at varying invader concentrations, which could then be used to determine $k_{\text{dissociation}}$ and $k_{\text{displacement}}$ through linear fitting.

Fig. S5 | Kinetic curves for displacing CP duplex using varying concentrations of invader strand (I) in the presence of varying concentrations of SG.

Fig. S6 | Observed rate constant k_{obs} determined at varying concentrations of SG from $0 \mu M$ to $1.25 \mu M$. The linear regression fitting was then used to determine $k_{dissociation}$ and $k_{displacement}$.

Fig. S7 | Dissociative rate constant $k_{dissociative}$ and strand displacement rate constant $k_{displacement}$ were plotted as a function of binder (SYBR Green-I) concentration. Changes in rate constants were highly consistent (bottom) with BIND profile, which confirmed the proposed reaction pathways at varying binder concentrations.

Comment-2: The way the Results section is written can lead to confusion because of insufficient experimental details described in this section. I suggest the authors clarify at the beginning of this section how the assay was set up before presenting data and discussing the results. Details need to be covered include the sequence, concentration, order/timing of addition of the DNA species and the exact buffer condition. I am also puzzled by the 1 M EDTA in the 1x TE buffer (in Supplementary Experimental Section).

Response-2: We thank the reviewer for the suggestion. We added a cartoon in the revised Supplementary Material to better illustrate the workflow of BIND and revised the manuscript to explain how the assay was set. There is a typo in the Supplementary Experimental Section. The concentration of EDTA was 1 mM rather than 1 M.

Revision-2: Supplementary Fig. S1 and corresponding discussion was added in the revised manuscript to better illustrate the workflow of BIND.

Supplementary Fig. S1 | Schematic illustration of the workflow for performing BIND reaction.

Comment-3: The authors argues that charge is one of the two determinants of whether a DNA-binder promotes the strand displacement after saturating the pre-formed DNA duplex. This explanation is not quite satisfying. One has to ask why charge is important for promoting strand displacement on a binder-stabilized dsDNA? The neutral molecules have $K_d > 100$ nM. Calling them "strong binder", a name also used for molecules with $K_d < 10$ nM, is hard to justify. Perhaps this is another place where the authors may want to examine the kinetics of the strand displacement reactions.

Response-3: In our experiments, we found that both specific binding and charge were critically needed to initiate a strand displacement through S_N2 reaction pathway. We reason that the charge is critical because it is needed to neutralize the strong charge repulsion of the CPI complex upon toehold docking. Despite the neutral binder may strengthen the hydrogen bonding at the toehold domain upon toehold docking, it could not help neutralize dense negative charges of the CPI complex. Here, we generally consider binders with K_d values lower than 1 μ M to be strong binder. We added this explanation in the

revised manuscript.

Comment-4: Out of the 20 small molecules, 4 of them did not return measurable K_d or n (Fig. S6). I commend the authors for including these data, but at the same time, urge them to discuss these phenomena.

Response-4: We did not report the K_d and n for four binders, including DAPP, RuPhen3Cl2, Crystal Violet and Thioflavin T. This is because these four binders were very weak binders, and they did not show the featured signal attenuation at the S_{N1} domain. As K_d and n were determined using data at the S_{N1} domain, we were able to determine the affinities of those four binders. To clarify the process for determining K_d and n , we added a workflow in the revised manuscript as Supplementary Fig. S11.

Revision-4: Supplementary Figure S11 was added into the revised manuscript to detail the process of data fitting for determining K_d and n .

Fig. S11 | Schematic illustration the workflow and mathematical transformation to determine the K_d and n of varying binders using BIND. The fractional occupancy from 30% and 80% was determined to be the optimal range for determining critical binding parameters, including dissociation constants, and binding site sizes.

Minor points:

Comment-5: The term "sequence preference" and "sequence selectivity" were used many times, leaving the impression that the DNA binders were systematically evaluated against a comprehensive

library of DNA sequences. However, only two 5-bp long DNA sequences were tested. So, at best, the data reflect the DNA binder's selectivity for GC-rich sequences. This needs to be clarified throughout the manuscript.

Response-5: We agree that the sequence selectivity was mainly selectivity to GC- or AT-rich sequences. We have clarified this in the revised manuscript.

Comment-6: What are the color codes used for the DNA binder names in Fig. 3. Why are there 4 groups?

Response-6: There is no specific meaning for the color codes. They were used just to help distinguish different binders.

Comment-7: Line 145, Fig. 4j and Fig. 4k should probably be Fig. 2j and Fig. 2k.

Response-7: Corrected as suggested by the reviewer.

REVIEWER COMMENTS

Reviewer #1 (Remarks to the Author):

The revised version of manuscript is greatly improved. I suggest the acceptance after minor revisions:

- 1. "Enabling programmable dynamic DNA nanotechnology" sounds a little wider than the demonstrate functions in this paper. A more focused title is preferred.**
- 2. In Fig. S4 to Fig. S6, the authors should clarify the fitting method to convert the scatter plots into kinetic curves and how they derived the $k_{\text{dissociative}}$ and $k_{\text{displacement}}$ from these kinetic curves.**
- 3. How to interpret the convex or concave shape of BIND curves in the SN1 region?**

Reviewer #2 (Remarks to the Author):

I have reviewed the revised manuscript entitled "Enabling programmable dynamic DNA nanotechnology using small-molecule DNA binders" submitted to Nature Communications. I would like to commend the authors for their efforts in addressing the concerns raised during the initial review process.

In the response letter and revised version, the authors have provided satisfactory and well-reasoned responses to all the questions and comments I had previously raised. The changes made have improved the clarity of the manuscript, and the additional data and analyses provided further support their conclusions.

Having evaluated the revisions and the authors' responses, I am pleased to recommend the acceptance of this manuscript for publication in Nature Communications.

Reviewer #3 (Remarks to the Author):

My concerns during the initial review have been adequately addressed by the revised manuscript. I recommend publication with the following comments.

I would like the authors to write about the new kinetic study in the main text Methods as well as in S2. Theoretical Modeling and Fitting, including details such as how the equation of $k_{\text{obs}} = k_{\text{dissociative}} + k_{\text{displacement}}[\text{Inv}]$ was established and how observed rate constant was determined from the kinetic curves. Additionally, in figure legends of Fig. S4 and S5, the meaning of the red curves should be clarified (e.g. fitting curve of exponential decay).

Response Letter

We thank all reviewers for their insightful comments and suggestions. We have thoroughly revised our manuscript accordingly. A point-by-point response is provided as follows:

Reviewer #1

The revised version of manuscript is greatly improved. I suggest the acceptance after minor revisions:

Comment-1. "Enabling programmable dynamic DNA nanotechnology" sounds a little wider than the demonstrate functions in this paper. A more focused title is preferred.

Response-1: Title was modified as suggested by the reviewer.

Revision-1: Enabling programmable dynamic DNA chemistry using small-molecule DNA binders

Comment-2. In Fig. S4 to Fig. S6, the authors should clarify the fitting method to convert the scatter plots into kinetic curves and how they derived the $k_{\text{dissociative}}$ and $k_{\text{displacement}}$ from these kinetic curves.

Response-2: This information was added to the captions of Fig. S4 to S6, the method section of the manuscript, as well as a kinetic model section in the supplementary materials.

Revision-2:

Method section:

Kinetic characterization of S_{N1} and S_{N2} pathways in BIND

Kinetics of BIND was investigated using a strand-exchange approach previously described by Reynaldo *et al.*²⁶ Briefly, 20 nM CP duplex in the absence or presence of varying concentrations of SG-I was invaded using increasing concentration of I from 250 nM to 2 μ M. The fluorescence of the reaction mixture was measured in real-time using a BioTek Cytation 5 Multimode Microplate reader at a data acquisition rate of one data point per minute for a period of 1 hour. The excitation/emission wavelengths were set to 640 nm/675 nm. Raw fluorescence of each data point was normalized against

a positive control and a negative control as described above to determine the reaction yield. The observed rate constant was then calculated by fitting the normalized experimental data against the following equation: $[CP] = [CP]_0 \cdot \exp(-k_{obs} \cdot t)$. Rate constants $k_{dissociative}$ and $k_{displacement}$ were then determined through a linear regression against the theoretical model previously described by Reynaldo *et al.*:²⁶

$$k_{obs} = k_{dissociative} + k_{displacement} \cdot [I]$$

Supplementary materials section S2:

Kinetic Model

Kinetic model of BIND was established based on DNA strand-exchange model previously described by Reynaldo *et al.*,¹ where both dissociative (S_N1) and sequential displacement (S_N2) pathways were considered for a toehold-free DNA strand exchange reaction. In this work, the observed rate constant k_{obs} was determined to be $k_{dissociative} + k_{displacement} \cdot [Invader]$, where $k_{dissociative}$ and $k_{displacement}$ were individual reaction rate constants for dissociative and sequential displacement pathways, respectively. Based on this model, we established the kinetic model for BIND, where the reaction rate

$$\frac{d[CP]}{dt} = -k_{dissociative} \cdot [CP] - k_{displacement} \cdot [CP][I]$$

$$\frac{d[CP]}{dt} = -(k_{dissociative} + k_{displacement}[I])[CP]$$

If $[I]_0$ is in large excess:

$$\frac{d[CP]}{dt} \approx -k_{obs} \cdot [CP]$$

$$k_{obs} = k_{dissociative} + k_{displacement} \cdot [I]_0$$

therefore,

$$[CP] = [CP]_0 \cdot \exp(-k_{obs} \cdot t)$$

$$Yield_t = \exp(-k_{obs} \cdot t)$$

Fig. S4 | Schematic illustration of the workflow for investigating the kinetics of BIND. Kinetic curves were collected when invading CP duplex using increasing concentrations of invader I. The observed rate constant k_{obs} were determined at varying invader concentrations, which could then be used to determine $k_{dissociative}$ and $k_{displacement}$ through linear fitting. Briefly, at each invader concentration, the value of k_{obs} was fitted by the following function: $[CP]_t = [CP]_0 \cdot \exp(-k_{obs} \cdot t)$. A linear regression against the theoretical model was used to determine $k_{dissociation}$ and $k_{displacement}$ thereafter, $k_{obs} = k_{dissociative} + k_{displacement} \cdot [I]$. All fitting curves were shown as red lines.

Fig. S5 | Kinetic curves for displacing CP duplex using varying concentrations of invader strand (I) in the presence of varying concentrations of SG. Values of k_{obs} were determined using the fitting approach as described in Fig. S4. All fitting curves were shown as red lines.

Fig. S6 | Observed rate constant k_{obs} determined at varying concentrations of SG from 0 μ M to 1.25 μ M. The linear regression fitting as described in Fig. S4 was then used to determine $k_{dissociative}$ and $k_{displacement}$. All fitting curves were shown as red lines.

Comment-3. How to interpret the convex or concave shape of BIND curves in the SN1 region?

Response-3: The convex or concave shape of BIND curve is determined by the binding cooperativity between a given binder and DNA. For binders with no cooperativity, the BIND curve in SN1 region shows a representative concave shape. For binders with positive cooperativity (in the case of evagreen), the affinity for the binder increases with more binders bound. Therefore, the binding curve shifts from a concave shape to convex shape with increasing cooperativity factor.

Reviewer #3

My concerns during the initial review have been adequately addressed by the revised manuscript. I

recommend publication with the following comments.

Comment-1. I would like the authors to write about the new kinetic study in the main text Methods as well as in S2. Theoretical Modeling and Fitting, including details such as how the equation of $k_{obs} = k_{dissociative} + k_{displacement}[Inv]$ was established and how observed rate constant was determined from the kinetic curves. Additionally, in figure legends of Fig. S4 and S5, the meaning of the red curves should be clarified (e.g. fitting curve of exponential decay).

Response-1: This information was added to the captions of Fig. S4 to S6, the method section of the manuscript, as well as a kinetic model section in the supplementary materials.

Revision-1:

Method section:

Kinetic characterization of S_{N1} and S_{N2} pathways in BIND

Kinetics of BIND was investigated using a strand-exchange approach previously described by Reynaldo *et al.*²⁶ Briefly, 20 nM CP duplex in the absence or presence of varying concentrations of SG-I was invaded using increasing concentration of I from 250 nM to 2 μ M. The fluorescence of the reaction mixture was measured in real-time using a BioTek Cytation 5 Multimode Microplate reader at a data acquisition rate of one data point per minute for a period of 1 hour. The excitation/emission wavelengths were set to 640 nm/675 nm. Raw fluorescence of each data point was normalized against a positive control and a negative control as described above to determine the reaction yield. The observed rate constant was then calculated by fitting the normalized experimental data against the following equation: $[CP] = [CP]_0 \cdot \exp(-k_{obs} \cdot t)$. Rate constants $k_{dissociative}$ and $k_{displacement}$ were then determined through a linear regression against the theoretical model previously described by Reynaldo *et al.*²⁶

$$k_{obs} = k_{dissociative} + k_{displacement} \cdot [I]$$

Supplementary materials section S2:

Kinetic Model

Kinetic model of BIND was established based on DNA strand-exchange model previously described by Reynaldo *et al.*,¹ where both dissociative (S_N1) and sequential displacement (S_N2) pathways were considered for a toehold-free DNA strand exchange reaction. In this work, the observed rate constant k_{obs} was determined to be $k_{dissociative} + k_{displacement} \cdot [Invader]$, where $k_{dissociative}$ and $k_{displacement}$ were individual reaction rate constants for dissociative and sequential displacement pathways, respectively. Based on this model, we established the kinetic model for BIND, where the reaction rate

$$\frac{d[CP]}{dt} = -k_{dissociative} \cdot [CP] - k_{displacement} \cdot [CP][I]$$

$$\frac{d[CP]}{dt} = -(k_{dissociative} + k_{displacement}[I])[CP]$$

If $[I]_0$ is in large excess:

$$\frac{d[CP]}{dt} \approx -k_{obs} \cdot [CP]$$

$$k_{obs} = k_{dissociative} + k_{displacement} \cdot [I]_0$$

therefore,

$$[CP] = [CP]_0 \cdot \exp(-k_{obs} \cdot t)$$

$$Yield_t = \exp(-k_{obs} \cdot t)$$

Fig. S4 | Schematic illustration of the workflow for investigating the kinetics of BIND. Kinetic curves were collected when invading CP duplex using increasing concentrations of invader I. The observed rate constant k_{obs} were determined at varying invader concentrations, which could then be used to determine $k_{dissociative}$ and $k_{displacement}$ through linear fitting. Briefly, at each invader concentration, the value of k_{obs} was fitted by the following function: $[CP]_t = [CP]_0 \cdot \exp(-k_{obs} \cdot t)$

t) . A linear regression against the theoretical model was used to determine $k_{dissociation}$ and $k_{displacement}$ thereafter, $k_{obs} = k_{dissociative} + k_{displacement} \cdot [I]$. All fitting curves were shown as red lines.

Fig. S5 | Kinetic curves for displacing CP duplex using varying concentrations of invader strand (I) in the presence of varying concentrations of SG. Values of k_{obs} were determined using the fitting approach as described in Fig. S4. All fitting curves were shown as red lines.

Fig. S6 | Observed rate constant k_{obs} determined at varying concentrations of SG from 0 μ M to 1.25 μ M. The linear regression fitting as described in Fig. S4 was then used to determine $k_{dissociative}$ and $k_{displacement}$. All fitting curves were shown as red lines.

REVIEWERS' COMMENTS

Reviewer #1 (Remarks to the Author):

My concerns have been addressed in the revision. I support the acceptance of this paper.